# Local volume concentration, packing domains, and scaling properties of chromatin

**Marcelo A Carignano[1†], Martin Kroeger[2†], Luay M Almassalha[3†], Vasundhara Agrawal[1], Wing Shun Li[4], Emily M Pujadas-Liwag[1], Rikkert J Nap[1], Vadim Backman[1]\*, Igal Szleifer[1,5]\***

[1]Department of Biomedical Engineering, Northwestern University, Evanston, United States; [2]Magnetism and Interface Physics & Computational Polymer Physics, Department of Materials, ETH Zurich, Zurich, Switzerland; [3]Department of Gastroenterology and Hepatology, Northwestern Memorial Hospital, Evanston, United States; [4]Applied Physics Program, Northwestern University, Chicago, United States; [5]Department of Chemistry, Northwestern University, Evanston, United States

**\*For correspondence:**
v-backman@northwestern.edu (VB);
igalsz@northwestern.edu (IS)

[†]These authors contributed equally to this work

**Competing interest:** The authors declare that no competing interests exist.

## Abstract

We propose the Self Returning Excluded Volume (SR-EV) model for the structure of chromatin based on stochastic rules and physical interactions. The SR-EV *rules of return* generate conformationally defined domains observed by single-cell imaging techniques. From nucleosome to chromosome scales, the model captures the overall chromatin organization as a corrugated system, with dense and dilute regions alternating in a manner that resembles the mixing of two disordered bi-continuous phases. This particular organizational topology is a consequence of the multiplicity of interactions and processes occurring in the nuclei, and mimicked by the proposed return rules. Single configuration properties and ensemble averages show a robust agreement between theoretical and experimental results including chromatin volume concentration, contact probability, packing domain identification and size characterization, and packing scaling behavior. Model and experimental results suggest that there is an inherent chromatin organization regardless of the cell character and resistant to an external forcing such as RAD21 degradation.

## eLife assessment

The authors develop a self-returning self-avoiding polymer model of chromosome organization and show that their framework can recapitulate at the same time local density and large-scale contact structural properties observed experimentally by various technologies. The presented theoretical framework and the results are **valuable** for the community of modelers working on 3D genomics. The work provides **solid** evidence that such a framework can be used, is reliable in describing chromatin organization at multiple scales, and could represent an interesting alternative to standard molecular dynamics simulations of chromatin polymer models.

## Introduction

Chromatin is a complex macromolecular fiber that results from the assembly of DNA with histone and non-histone proteins to form the functional organization of the genome within the eukaryotic cell nucleus. That over 2-linear meters ($\sim 6 \times 10^9$ base pairs) is confined within human nuclei ranging between 5 and 10 μm in diameter while maintaining functionally relevant information creates a core dilemma that places a tension between efficiency of packing with information retention (*Annunziato,*

*2008*). Adding to this complexity are the rich heterogeneity of non-chromatin nuclear bodies, histone concentrations within normal cells, and chromosome copy number (and total DNA content) in malignant cells (*Clapier and Cairns, 2009*; *Tessarz and Kouzarides, 2014*; *Finn et al., 2019*; *Mansisidor and Risca, 2022*). Despite the profound degree of variability from cell-to-cell even within microscopically normal tissues (*Nagano et al., 2013*), the ensemble function of organs is maintained by facilitating the preferential activation of specific gene network patterns. In these contexts, describing chromatin as a stochasticaly evolving process with constraints appears to be an alternative, complementary approach to represent the regulatory processes that couple structure with function (*Sood and Misteli, 2022*).

Numerous polymer models of chromatin organization have been proposed to predict possible configurations (*Fujishiro and Sasai, 2022*; *Adame-Arana et al., 2023*; *Forte et al., 2023*; *Shi and Thirumalai, 2021*; *Tamm et al., 2015*; *Polovnikov et al., 2018*; *Mirny, 2011*). Many models have been motivated by the properties observed in Hi-C, with some recent studies interested in recapitulating the microphase separation observed microscopically (*Fujishiro and Sasai, 2022*; *Adame-Arana et al., 2023*). Heteropolymer models, where the monomers are partitioned into two groups, can achieve microphase separation by introducing attractive potentials between components of each group (e.g. 'b' monomers are attracted to 'b' but repulsed by 'a') (*Fujishiro and Sasai, 2022*; *Adame-Arana et al., 2023*). Likewise, introduction of spatially defined long-range loops to approximate cohesin-mediated loop extrusion can similarly *regulate* microporous structures (*Nuebler et al., 2018*). Existing homopolymer models that have been proposed either have limitations in their degree of coarse-graining (e.g. HIPPS(Hi-C-polymer-physics-structures) monomers are composed of 1200 bp, ~6 nucleosomes with a diameter of ~60 nm) or are variations on a random walk polymer (*Forte et al., 2023*; *Shi and Thirumalai, 2021*). Homopolymer models are unable to achieve the biphasic, porous states observed on ChromEM, configurations that reliably result in contact scaling ($S$) less than $-1$ at supranucleosome length scales ($10^5$ to $10^6$ bps) as is frequently observed in Hi-C (*Tamm et al., 2015*; *Polovnikov et al., 2018*; *Mirny, 2011*) while also producing the observed physiologic range of chromatin power-law mass-density organization (scaling exponent, $D$) that ranges between 2 and 3.

Regarding these last points, a fundamental issue results from solely using measures of connectivity, such as Hi-C, for polymer modeling of chromatin organization due to the inverse relationship in polymers between mass density and contact scaling that obey $M \propto r^D$ and $D \sim 3(-S)$. Thus, the widely observed $S < -1$ results in configurations with mass in excess of the volume capacity ($D > 3$). Likewise, a random walk polymer model can achieve the limiting cases of $D = 2$ (a polymer in a $\Theta$ solvent) and $D = 3$ (a random walk in a confined volume) but these cases limit the functional role of chromatin to facilitate enzymatic processes (RNA transcription, replication, repair) with nucleosome size monomers. For example, in $D = 2$, nuclear enzymes would be diffusing through very large nuclei with a fully accessible genome whereas in $D = 3$, there is scant accessible space resulting in exclusive molecular activity at the surface of the genome. $D$ ranging between 2 and 3 is not achievable with existing models while accounting for volume considerations. This range has functional consequences as it produces genomic configurations that will be inaccessible (domain centers), surfaces for enzymatic activity, and low-density spaces for molecular mobility.

There have been important efforts to model chromatin and a comprehensive review have been recently published (*Yildirim et al., 2022*). Many works are based on atomistic or a nearly atomistic approach addressing different processes involving DNA, histones, and other proteins (*Bishop, 2005*; *Eslami-Mossallam et al., 2016*; *Bowerman and Wereszczynski, 2016*; *Melters et al., 2019*; *Zhang et al., 2017*; *Freeman et al., 2014*; *Lequieu et al., 2016*; *Brandani, 2018*; *Lequieu et al., 2017*; *Lequieu et al., 2019*; *Li et al., 2023*; *Arya et al., 2006*; *Arya and Schlick, 2009*; *Dans et al., 2016*; *Jimenez-Useche et al., 2014*; *Norouzi and Zhurkin, 2015*; *Bajpai and Padinhateeri, 2020*; *Luque et al., 2014*; *Perišić et al., 2019*; *Bascom et al., 2019*; *Bascom et al., 2017*; *Wiese et al., 2019*). From the other end of the chromatin length scale the aim is to use experimental results, especially from high-throughput chromatin conformation capture (Hi-C) (*Lieberman-Aiden et al., 2009*), to guide polymer models simulations with especial characteristics that can replicate for example, contact patterns and loop extrusion process (*Banigan and Mirny, 2020*; *Barbieri et al., 2012*; *Brackley et al., 2017*; *Fudenberg et al., 2016*; *Nuebler et al., 2018*; *Rao et al., 2014*; *Sanborn et al., 2015*; *Chan and Rubinstein, 2023*; *Jost et al., 2014*). Many lines of evidence support the idea of chromatin configurations as a statistical assembly that produce functional organization. First, the overwhelming

majority of the genome does not code for proteins but has functional consequences at the level of regulating gene transcription. Second, Hi-C (*Lieberman-Aiden et al., 2009*) and similar techniques identify the presence of compartments, domains, and loops; however, these structures only become evident as distinct contact loci with millions of sequence measurements (*Szabo et al., 2019*; *Rajderkar et al., 2023*). Third, single-cell sequencing and in situ sequencing of normal tissue and malignancies has demonstrated profound heterogeneity in transcriptional patterns that were previously not appreciated under routine histological examination (*Finn et al., 2019*). Finally, ongoing methods investigating chromatins structure have shown that it is dynamically evolving even at the order of seconds to minutes (*Nagano et al., 2017*).

We present herein a minimal model based purely on molecular, physical, and statistical principles which (1) preserves the efficiency of chromatin packing, (2) produces the structural heterogeneity and population diversity observed experimentally, (3) retains the capacity for functionally relevant storage of genomic information across modalities, and (4) would be sensitive to variations in density present in clinically relevant contexts (i.e. ploidy and nuclear size are frequently varied in cancer and mammalian cells have a distribution of nuclear sizes that vary by tissue function). To produce this model, we began by *assuming* that there is an overall statistical rule governing the spatial organization of chromatin. Inspired by known features of genome organization, (1) nucleosomes are the base structure of the chromatin polymer, (2) long-range interactions arise from a plurality of mechanisms (loop extrusion, promoter–promoter interactions, promoter–enhancer interactions, and spatial confinement), and (3) the volume fraction of chromatin depends on genomic content coupled with nuclear size which therefore varies in different tissues and states. We show by representing these processes from the interplay of stochastically occurring low-frequency, large extrusion returns (stochastic-returns) probabilistically from multiple processes in the context that monomers occupy physical space (excluded volume) that the missing features of chromatin polymer modeling are obtained. We demonstrate first that this model recapitulates the ground-truth structure of chromatin on methods that measure structure (chromatin electron microscopy, partial wave spectroscopic [PWS] microscopy) and connectivity (Hi-C). The findings from the model address the deficiencies occurring in existing literature: the biphasic structures on chromatin electron microscopy is observed, scaling and space-filling properties are preserved, and the expected population heterogeneity arises de novo from stochastically produced configurations. With a minimal model depending on just two parameters, we demonstrate the production of irregular fiber assembles with a radius of ~60 nm while producing the average nuclear density of 2030%.

In agreement with chromatin scanning transmission electron microscopy (ChromSTEM) and many other experimental methods, we find that genomic structure has a characteristic radial dependency that can be interpreted in terms of a power-law with exponent $D$. Comparing Self Returning Excluded Volume (SR-EV) to live-cell PWS microscopy, we demonstrate that the diversity in chromatin configurations observed on SR-EV corresponds with experimental observations. We then test the distinct roles that long-range returns and excluded volume have on structure using an auxin-inducible degron RAD21 cell line, allowing the depletion of a core component of the cohesin complex that can be quantitatively tested by PWS and ChromSTEM microscopy. In our model, the long-range steps arise from a confluence of processes and the inhibition of one of these processes like for example cohesin-mediated loop extrusion has a limited effect on chromatin packing. Remarkably, our model demonstrates that upon RAD21 depletion, only ~20% decrease in the number of observed domains, with the remaining domains largely unaffected at the level of their size, density, and $D$; results recapitulated directly on ChromSTEM imaging. Furthermore, depletion of RAD21 is predicted to have a minor effect on the diversity of chromatin configurations, a finding again confirmed with live-cell PWS microscopy. Finally, we show that excluded volume results in a non-linear, monotonic relationship between power-law organization and local density that plateaus near $D$ of 2.8, predictions observed with and without RAD21 present in ChromSTEM.

The structures predicted by our model display a porosity that result from the alternation of high- and low-density regions. The envelope of the high-density regions could be regarded as the separating interphase of a bi-continuous system that is a topological scenario that favors extensive mobility of proteins, mRNA, and other free crowders while providing a large accessible surface area of chromatin. The contact probability, calculated as an ensemble average, shows a good agreement with Hi-C results displaying a transition between intra- and inter-domain regimes. The intra-domain contact

probability scales with an exponent $S > -1$, while the inter-domain one scales with an exponent $S \sim -1$. As such, this work introduces the basis for a statistical representation of the genome structure.

## Results

### A minimal model for chromatin conformations

The SR-EV model for chromatin is derived from the Self Returning Random Walk (SRRW) model that was recently introduced by this group (*Huang et al., 2020*). Here, we review the SRRW model and then we introduce the modifications that lead to the SR-EV model.

The SRRW model is essentially a random walk with specific rules introduced to capture statistical features of chromatin organization as revealed by experiments. At each step in the SRRW generation there are two possibilities: (1) Perform a forward jump or (2) Return over the previous step to the previous position. The probability $P_R$ for a return step is given by

$$P_R(U_0) = \frac{U_0^{-\alpha}}{\alpha}$$

(1)

Here, $U_0$ is the length of the last step along the backbone over which the walk may return. The folding parameter $\alpha > 1$ controls the number of returns. If the SRRW does not continue with a return step, it must continue with a forward jump. The new forward jump is chosen with an random direction and with a length $U_1$ given by the following probability distribution function (pdf)

$$P_J(U_1 > 1) = \frac{\alpha + 1}{U_1^{\alpha+2}}$$

(2)

We will generally refer to *Equations 1 and 2* as the *return rules* of the SR-EV model. There is a minimum size for the forward jumps that also defines the unit of length in the model. The succession of forward jumps and return steps leads to a structure than can be regarded as a linear backbone

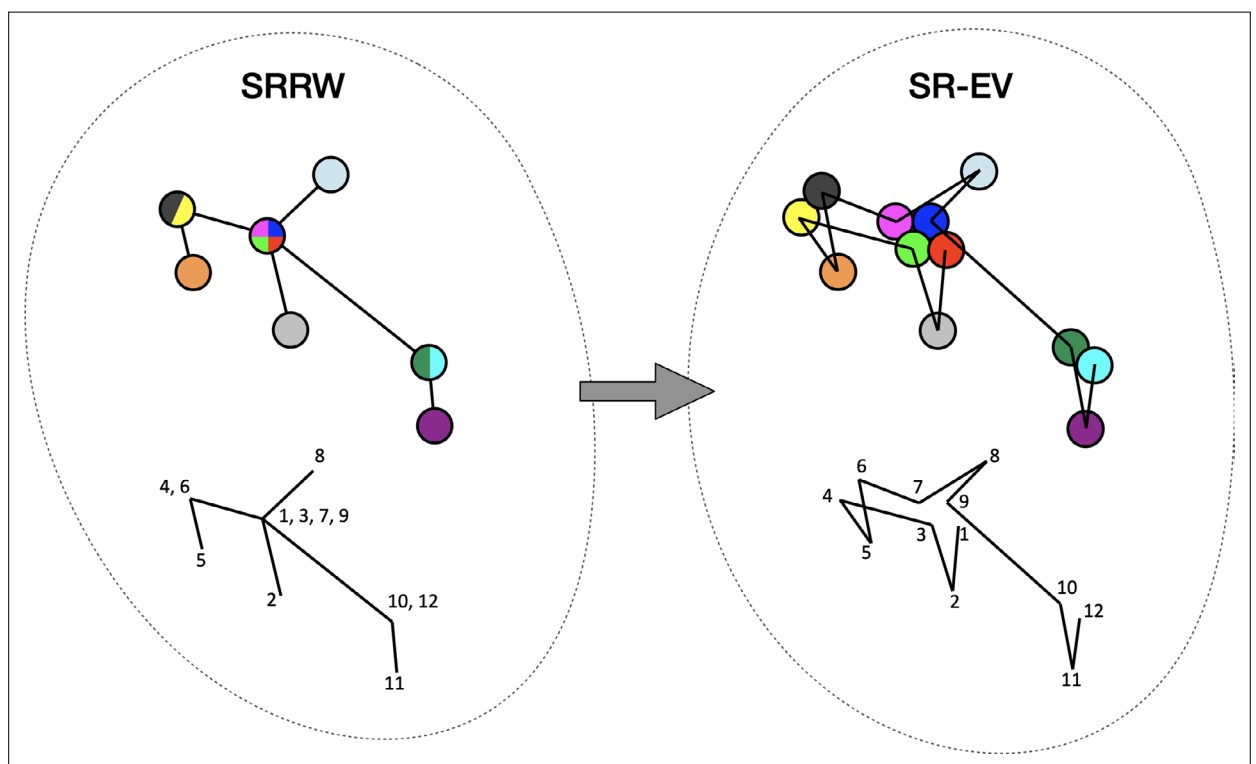

**Figure 1.** Schematic representation of the conversion process from Self Returning Random Walk (SRRW) to Self Returning Excluded Volume (SR-EV). The SRRW configurational motif hides the overlap of several beads in a molecule that has the structure of a branching polymer. By the introduction of excluded volume in SR-EV, the overlapping beads separate to form a cluster and a linear molecule.

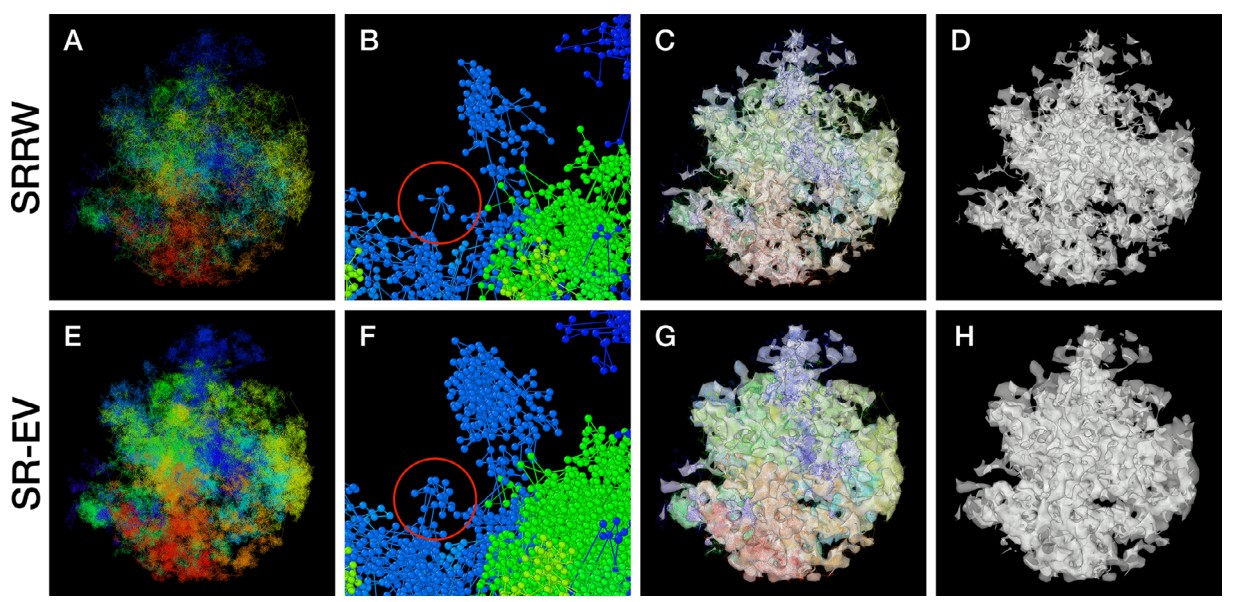

**Figure 2.** Example Self Returning Random Walk (SRRW) and Self Returning Excluded Volume (SR-EV) configurations. The top rows are for the SRRW case, and bottom row corresponds to the associated SR-EV configuration. (**A**) and (**E**) represent the bonds of the full configurations and show that while SR-EV looks denser than the SRRW case the overall structure is preserved upon removal of the original overlaps. (**B**) and (**F**) correspond to the same small portion of the conformation and shows SR-EV having many more beads than SRRW due to the excluded volume between beads. The red circles explicitly highlight a structural motif that in SRRW is a central bead with 7 bonds branching out (a sequence of seven consecutive jump and returns steps) that transform to 15 linearly connecting beads forming a cluster. (**C**) and (**G**) display the chromatin conformations wrapped by a tight mesh suggesting the separation between a chromatin-rich and a chromatin-depleted regions, the latter being the space that free crowders could easily occupy. (**D**) and (**H**) show the bare interface between the two regions that resembles the interface dividing two bi-continuous phases and also clearly expose the difference between SRRW and SR-EV.

The online version of this article includes the following video(s) for figure 2:

**Figure 2—video 1.** Self Returning Excluded Volume (SR-EV) configuration represented as beads and sticks, wrapped in a mesh envelope that separates the dense regions from the nearly empty regions of the configuration.

https://elifesciences.org/articles/97604/figures#fig2video1

**Figure 2—video 2.** Representation of the same system of *Figure 2—video 1*, but in this case it contains only the wrapping mesh that resembles the interface between two disordered bi-continuous phases.

https://elifesciences.org/articles/97604/figures#fig2video2

with tree-like branches along its length, with the branching points representing overlaps created by the return steps. In addition to the return probability and pdf defined above, the SRRW generation algorithm (contained in Appendix 1) includes a local cutoff to avoid unrealistically long steps and a spherical global cutoff to contain the configuration. The global cutoff is applied during the generation of the conformation and is measured from the center of mass of the already-generated steps.

By construction, since the SRRW includes returns over the previous steps, it contains a large number of overlaps. For $\alpha$ = 1.10, 1.15, and 1.20 the number of returns is 48.7%, 47.5%, and 46.2% of the total number of steps, respectively. Therefore, as a representation of a physical system, such as chromatin, the SRRW has two important drawbacks: (1) the conformations violate the principle of excluded volume and (2) it is not a linear polymer. In order to recover these two physical properties we extended the SRRW to develop the SR-EV model. In this new method, the overlapping points are transformed into connected clusters of beads that explicitly represent a linear chain, as shown on the scheme displayed in *Figure 1*. The method that we employ to remove overlaps is a low-temperature-controlled molecular dynamics simulation using a soft repulsive interaction potential between initially overlapping beads, that is terminated as soon as *all* overlaps have been resolved, as described in the Appendix 1. An example of an SRRW configuration and its corresponding SR-EV are displayed in *Figure 2A and E*, respectively. *Figure 2B, F* represents a small region on the periphery of the configuration and exemplifies how structures formed by a sequence of forward and returns steps expands

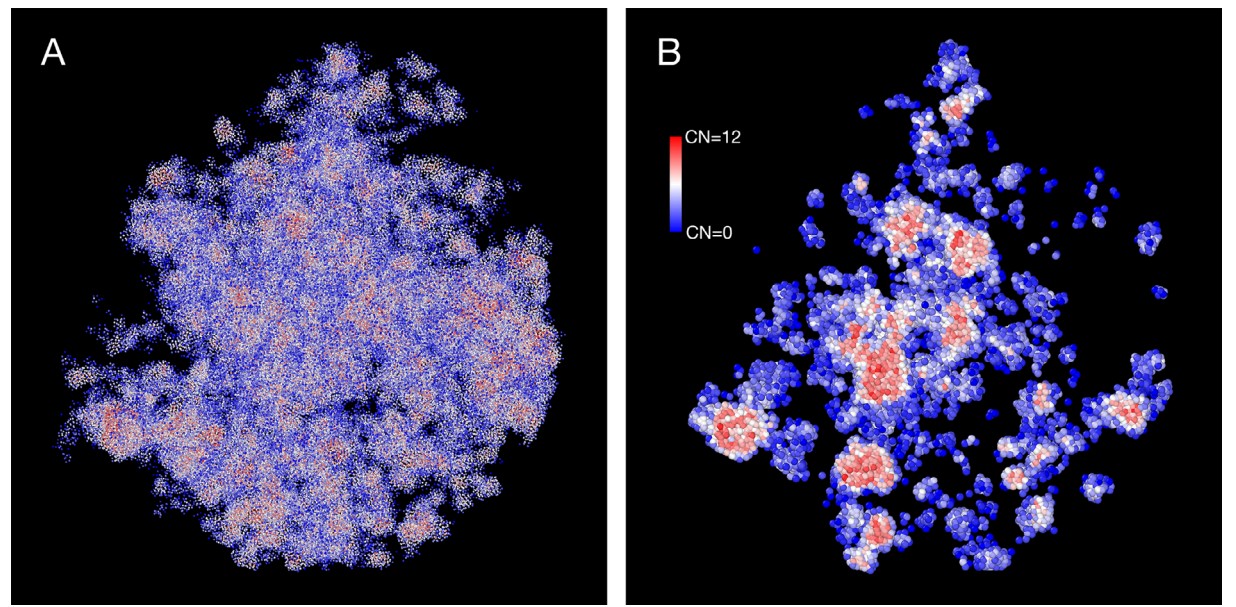

**Figure 3.** Packing domains and nucleosome accessibility. Same Self Returning Excluded Volume (SR-EV) configuration displayed in *Figure 2*, but colored by the coordination number of each nucleosome. (**A**) Full configuration reveals the spacial dispersity of packing domains in red, consistent with heterochromatic region, intercalated with low coordinated, accesible regions. (**B**) 50 nm slab cut at the center of the configuration displaying details of the system heterogeneity and transition from packing domains to the intermediate, low coordinated, region. Note the white nucleosomes (coordinations number [CN] ~ 6) at the periphery of the packing domains.

to a larger cluster after including excluded volume interactions. The porosity of the structure is also affected by the excluded volume introduced in SR-EV.

The density heterogeneity displayed by the SR-EV configurations can be analyzed in terms of the accessibility. One way to reveal this accessibility is by calculating the coordinations number (CN) for each nucleosome, using a coordination radius of 11.5 nm, along the SR-EV configuration. CN values range from 0 for an isolated nucleosome to 12 for a nucleosome immersed in a packing domain. In *Figure 3*, we show the SR-EV configuration shown in *Figure 2*, but colored according to CN. CN can be also considered as a measure to discriminate heterochromatin (red) and euchromatin (blue). *Figure 3A* shows how the density inhomogeneity is coupled to different CN, with high CN represented in red and low CN represented in blue. *Figure 3B* shows a 50-nm thick slab obtained from the same configuration that clearly shows the nucleosomes at the center of each packing domain are almost completely inaccesible, while those outside are open and accessible. It is also clear that the surface of the packing domains is characterized by nearly white nucleosomes, i.e. coordinated toward the center of the domain and open in the opposite direction.

For this work, we adopted a *unit length* of 10 nm, similar to the diameter of a nucleosome (*Maeshima et al., 2014*). Therefore, each bead of the model chromatin represents a nucleosome. The spherical global cutoff was set to $R_c = 650$ nm. From the resulting conformations, we can cut slabs spanning well over 1 μm in cross section. Excluded volume was introduced by imposing a non-overlap radius of $r_o = 4.9$ nm between all the beads of the SR-EV model. With these quantities, we defined the overall average volume fraction as $\phi = N(r_o/R_c)^3$, with $N$ the number of beads in the chromatin model chain. We considered four different volume fractions $\phi = 0.08, 0.12, 0.16$, and $0.20$, which correspond to $N = 186{,}741, 280{,}112, 373{,}483$, and $466{,}854$, respectively. Each one of these four average volume fractions was studied with three different folding parameters $\alpha = 1.10, 1.15$, and $1.20$. SR-EV configurations, as we present them in this work, are associated to the structure of a single chromosome. Therefore, all the analysis that follows is done on the structure of a single chromosome system. For each combination of $\phi$ and $\alpha$, we created an ensemble of 1000 different chromatin configurations. In order to introduce the genomic distance along the SR-EV configuration we assign 147 base pairs to each nucleosome, representing the length of DNA wrapping the histone octamers. Considering that the effective bead diameter is 9.8 nm, the average distance between adjacent base pairs in the DNA

**Table 1.** Linker DNA mean value for the 12 $\Phi$, $\alpha$ studied combinations.

The folding parameter $\alpha$ controls the return rules, **Equations 1 and 2**. $N$ is the total number of nucleosomes represented in the model, which is related to the overall volume fraction with $r_o$ representing the radius of the nucleosomes and $R_c$ the global spherical cutoff. The average number of DNA base pairs per model nucleosome, including the linker DNA, is 186.6.

**Mean value of linker DNA length (bp)**

| | | $\alpha$ | | |
|---|---|---|---|---|
| $\varphi$ | N | 1.10 | 1.15 | 1.20 |
| 0.08 | 186741 | 40.8 | 38.0 | 36.3 |
| 0.12 | 280112 | 41.8 | 38.6 | 36.6 |
| 0.16 | 373483 | 44.4 | 39.4 | 37.4 |
| 0.20 | 466854 | 43.2 | 42.0 | 36.9 |

double helix, and the model bonds $U_i$ that are larger than 10 nm, we assign the number of base pairs in the linker DNA as the nearest integer of $(U_i - 9.8\,\mathrm{nm})/(0.34\,\mathrm{nm})$. In **Table 1**, we summarize the 12 studied cases with the resulting mean value for the length, in base pairs, of the linker DNA between nucleosomes that slightly depends on $\phi$ and $\alpha$. The overall average length of the linker DNA sections is 39.6 base pairs and with values of 36.3 and 44.4 for the two extreme cases. We must remark that the predicted DNA length between histone octamers agrees with the widely reported values (**Beshnova et al., 2014**; **Wang et al., 2021**; **Lequieu et al., 2019**; **Li et al., 2023**; **Zhurkin and Norouzi, 2021**). Finally, and in order to correlate our work with experimental examples, the longest simulated chromatin corresponds to $88 \times 10^6$ base pairs, which is approximately the size of human chromosome 16.

## SR-EV reproduces the biphasic chromatin structures observed in ChromSTEM imaging

In order to start assessing whether the SR-EV model produces realistic configurations of chromatin it is necessary to bring the model to a representation similar to the results of imaging experiments. For example, ChromSTEM captures the chromatin density from a slab of 100 nm thickness. Then, we cut a similar slab from an SR-EV configuration and transform the point coordinates of the model

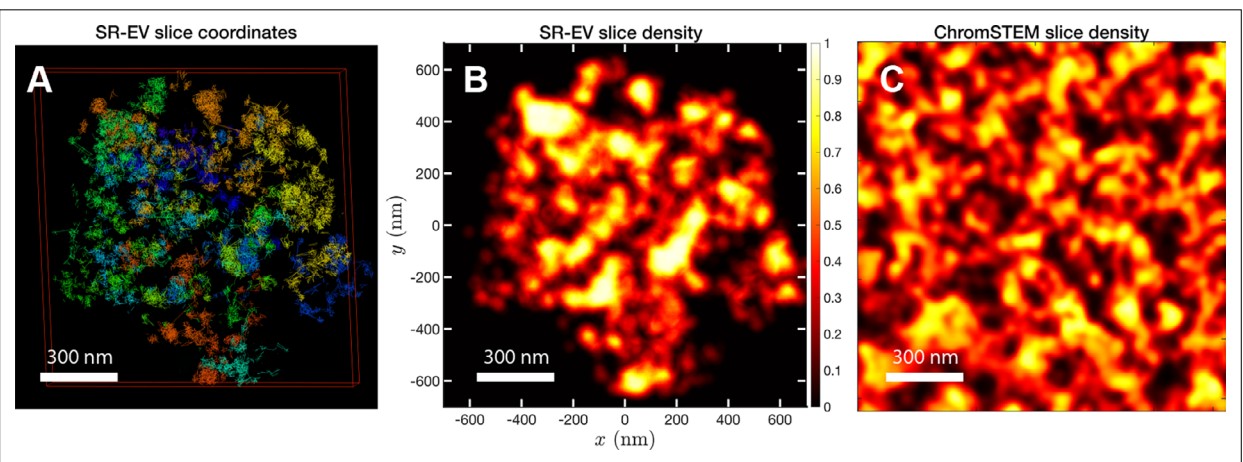

**Figure 4.** SR-EV and experimental slab images. (**A**) representation of a 100-nm slab cut at the center of an SR-VE conformation obtained with $\phi = 0.16$ and . (**B**) 2D chromatin density corresponding to coordinates of panel (**A**). (**C**) Chromatin scanning transmission electron microscopy (ChromSTEM) 2D chromatin density obtained from a 100-nm slab of a A549 cell. The 2D density color scale is the same for (**B, C**), and the density is normalized to its highest value in each image.

The online version of this article includes the following video for figure 4:

**Figure 4—video 1.** Stack of images from a conformation obtained with $\alpha = 1.20$ and $\phi = 0.12$.

https://elifesciences.org/articles/97604/figures#fig4video1

nucleosomes to a two dimensional density that considers the nucleosomes volume. In *Figure 4A*, we show a representation of an SR-EV configuration as it result from the model and in *Figure 4B* the collapsed two dimensional density as a colormap highlighting the porosity of the model and the emergence of chromatin packing domains. In *Figure 4C*, we show a ChromSTEM image for A549 cell. Since our SR-EV structures represent a single chromosome, it does not cover the full field of view of 1300 nm × 1300 nm that can be appreciated in the experimental image. However, the qualitative resemblance of the theoretical and experimental chromatin densities is stunning. The quantitative characterization of the model and its agreement with experimental results is analyzed below.

SR-EV is a non-homogeneous polymer model. The only physical interactions present in the model are the *connectivity*, the *excluded volume*, and the *confinement* that, together with the *return rules* induce the formation of granular structures, or packing domains, with local density variations. This granularity can be qualitatively visualized by wrapping a mesh around the chromatin conformation, as shown in *Figure 2G, H*. Rotating versions of *Figure 2G, H* are included in *Figure 2—video 1* and *Figure 2—video 2*. It is worth noting that this representation is qualitatively similar to *Figure 4*, panels E, F, and G from *Ou et al., 2017*. At first glance, the wrapping interface between the region denser in chromatin and the region almost empty of chromatin resembles the dividing interface between two disordered bi-continuous liquid phases (*Walker et al., 2014*). We find this outcome from the SR-EV model quite interesting in view of recent claims that liquid–liquid phase separation could be related to heterochromatin and euchromatin segregation, and that chromatin domains have a liquid character (*Chen et al., 2022*; *Itoh et al., 2021*). Moreover, the bi-continuous topology offers two important functional advantages: First, the interface offers a very large surface area exposing the a significant fraction of the genome and second, the continuity of the dilute phase allows for the migration of free crowders (including proteins, transcription agents, mRNA, etc.) to any region in the nucleus.

## SR-EV demonstrates that genome connectivity decouples from domain structure

The granularity of chromatin manifest itself in the polymeric properties of the model. Chromatin is a special type of polymer, and requires a careful analysis. The scaling relationship between the end-to-end distance and the polymer contour length, in this case the genomic distance, cannot be described in general with a single power-law relationship, i.e. a single Flory exponent, as it is the case for synthetic polymers. In *Figure 5A*, we display the ensemble averaged end-to-end distance, $\langle R^2(n)\rangle^{0.5}$ as a function of the genomic distance $n$. All the studied cases are included in the plot, but they coalesce in three distinct groups according to the folding parameter $\alpha$ and with almost no effect of the overall volume fraction. The figure also shows a transition occurring for $n \sim 4\times10^4$ base pairs, from a local or intra-domain regime that corresponds with distances up to 100 nm, to a long-range

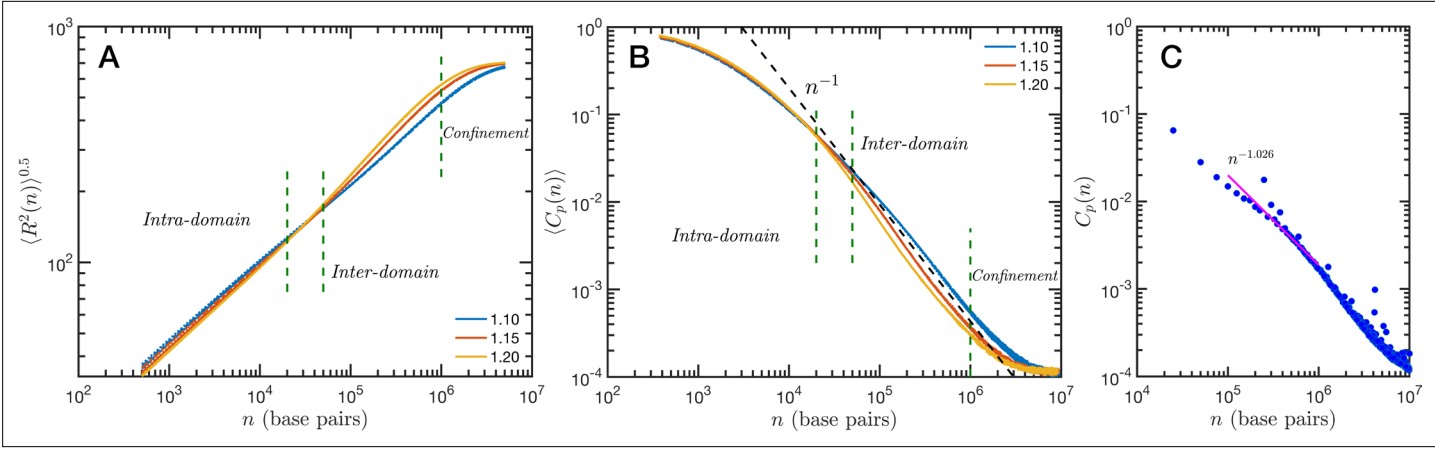

**Figure 5.** Theoretical and experimental polymeric properties of chromatin. Self Returning Excluded Volume (SR-EV) ensemble average of (**A**) end-to-end distance and (**B**) contact probability a as a function of the genomic distance for all simulated conditions, as described in *Table 1*. The crossover between short distance intra-domain and long distance inter-domain regimes is explicitly indicated, as well as the confinement effect at longer distances. Notice that on these two panels there are four lines per $\alpha$ value, while $\alpha \in \{1.10, 1.15, 1.20\}$. (**C**) Experimental (Hi-C) contact probability for chromosome 1 of HCT-116 cells showing quantitative agreement with the theoretical results.

or inter-domain one. The Flory exponent in the intra-domain regime (0.342, 0.347, and 0.354 for $\alpha = 1.10$, 1.15, and 1.20, respectively) is consistent with a nearly space-filling cluster and slightly smaller than in the inter-domain regime (0.353, 0.394, and 0.396). For $n$ values larger than $10^6$ the curves level off due to the effect of the spherical confinement. The analysis can also be applied to the ensemble average contact probability, $\langle C_p(n) \rangle$, which is defined as the probability for two base pairs, separated along the polymer by a genomic distance $n$, of being in contact with each other (or being at a distance smaller than a cutoff). In *Figure 5B*, we display $\langle C_p(n) \rangle$ for all studied cases, using a cutoff distance of 35 nm. We see in this figure that the contact curves depend only marginally on volume fraction as the four distinct cases for each $\alpha$ are nearly indistinguishable. Thus, this indicates that measures of connectivity observed in Hi-C would not depend on nuclear volume concentrations. This finding is in strong agreement with the results reported in *Liu and Dekker, 2022* where expansion and contraction of isolated nuclei has minimal effects on contact scaling, $S$. As in the end-to-end distance, in $\langle C_p(n) \rangle$ we can also distinguish a transition between intra- and inter-domain regimes. In general, the slope $S$ of $\langle C_p(n) \rangle$ in log–log representation is larger than $-1$ in the inter-domain regime, and fluctuate around $-1$ for inter-domain genomic distances. *Figure 5C* shows the contact probability determined from Hi-C experiments. The blue dots correspond to chromosome 1 of HCT-116 cells and the behavior between $10^5$ and $10^6$ base pairs is well described by a slope $S$ very close to $-1$. The experimental data also show a change at intermediate separations. It is important to note to the agreement is relatively good even in quantitative terms, with the transition occurring at similar genomic distance and value of $C_p(n)$. Since the model does not have a genomic identity or any specific architectural modifiers (e.g. CTCF(CCCTC-binding factor) and/or cohesin), the contact probability curves do not represent a particular cell or chromosome. We must mention that the other chromosomes from the HCT-116 cells have a qualitatively similar contact probability, with a power-law fitting having slopes $S$ varying from $-0.85$ to $-1.10$, depending the case.

## Chromatin volume concentrations couple with long-range returns to determine 3D structure

The heterogeneous character of chromatin revealed by experiments is captured, as we have qualitatively shown above, by the SR-EV model. A straightforward characterization of this heterogeneity is the distribution of local volume fraction calculated with a probing volume of adequate size. In the language common in chromatin experiments, this volume fraction is referred to as the chromatin volume concentration (CVC) and the probing volume is, for example, a cube with an edge of 120 nm. Using electron microscopy and tomography techniques (ChromEMT), the group of Dr Clodagh O'Shea (*Ou et al., 2017*) reconstructed the conformation of chromatin on a 120-nm thick slab with an area of 963 nm × 963 nm, which allowed them to measure the CVC distribution using a 8 × 8 × 1 grid with cubic cells of 120 nm edge size. To calculate the CVC from the SR-EV configuration ensembles we followed the same methodology employed in the experiments. Since we have the full 3D structure of the model chromatin we are not restricted to a slab, then we used a 6 × 6 × 6 cubic grid of $(120 \text{ nm})^3$ probing volumes. Moreover, our results represent ensemble averages over the populations of 1000 replicates for each of the $\Phi$ and $\alpha$ combinations. The results for each case are summarized in *Figure 6* revealing that both SR-EV parameters, $\Phi$ and $\alpha$, are important in determining the CVC distributions. We see that overall the volume fraction take values up to 0.6, which is consistent with our model representing the nucleosomes as spheres that can achieve a maximum volume fraction of 0.74 as a crystal and 0.64 in the jamming limit (*Jin and Yoshino, 2021*). The peak of the CVC distribution increases as the overall volume fraction $\alpha$ increases. The recent ChromEMT results reveal a CVC distribution covering a nearly identical range to our SR-EV results. Comparing with the experimental results, for the lowest overall volume fraction the distribution has an excessive proportion of low-density regions. Consequently, although we show that all regimes will result in domain formation with $\alpha = 1.15$ and $\phi = 0.16$ being the closest to what is observed in A549 cells (*Figure 7C*), this would indicate that the variation in chromatin density that arises in mammalian cells would be predicted to have distinct functional consequences that would not be captured by connectivity. As we show below, chromatin packing domain organization will be weakly and inversely related to the probability of return events but strongly associated with the local volume concentrations.

Since the CVC is a measure using a relative large probing volume its distribution with values ranging from 0 to 0.6 may be achieved by a (dynamic) smooth continuous modulation of the chromatin density

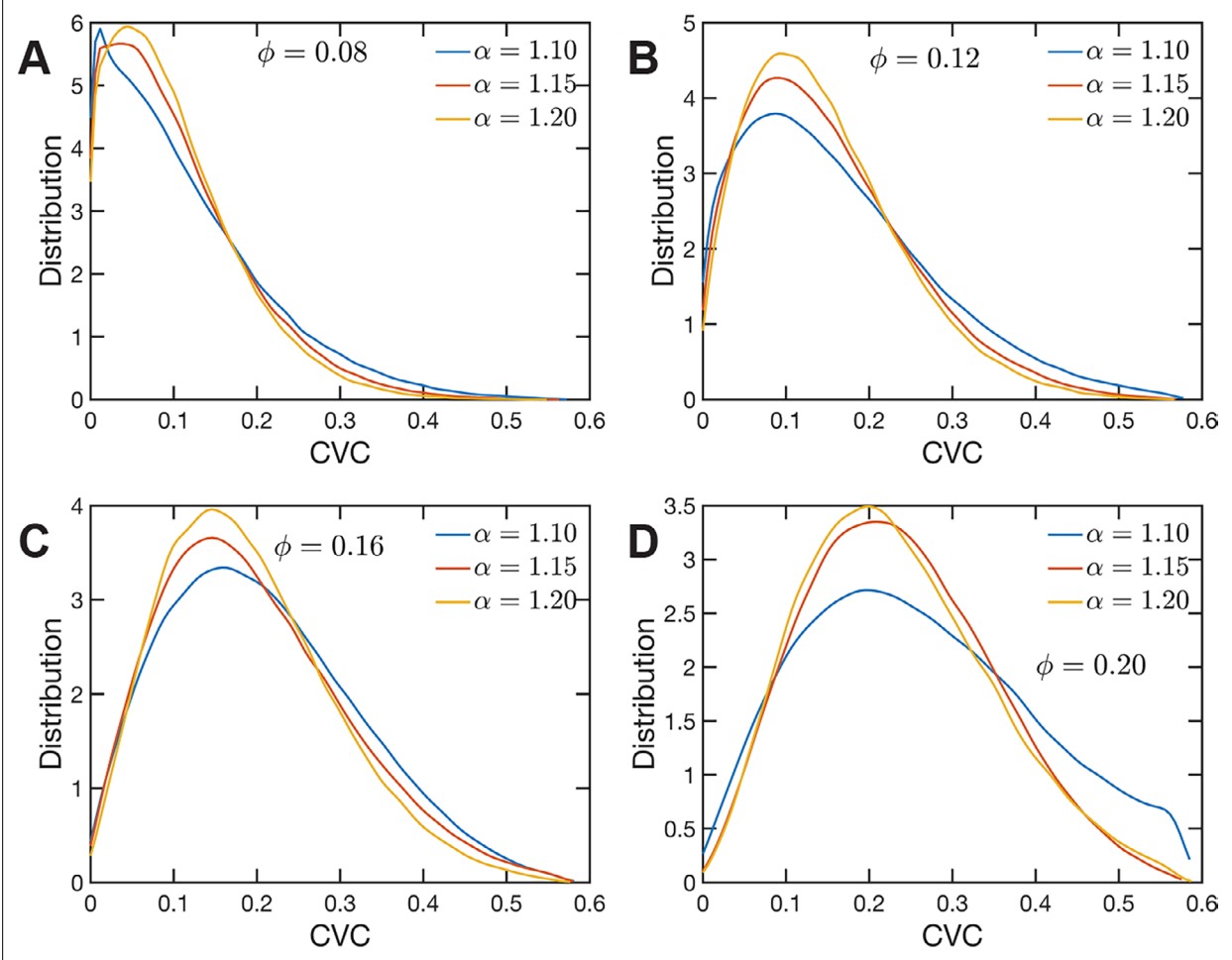

**Figure 6.** Chromatin volume concentration (CVC) for (**A**) $\phi = 0.08$, (**B**) $\phi = 0.12$, (**C**) $\phi = 0.16$ and (**D**) $\phi = 0.20$ and $\alpha \in \{1.10, 1.15, 1.20\}$. The results for $\phi = 0.20$, $\alpha = 1.15$ are the closest to the experimental findings of *Ou et al., 2017*. $\phi = 0.08$ produce CVC distributions with a much larger contribution of low-density regions, and $\phi = 0.20$, $\alpha = 1.10$ over enhance the high-density regions.

or by a (also dynamic) mixing of distinct high- and low-density regions. The latter scheme gives rise to the concept of packing domains, as it has been recently proposed from the analysis of imaging experiments (*Li et al., 2022*; *Li et al., 2021*; *Miron et al., 2020*). The formation of domains is also consistent with the possibility of a microphase separation process dynamically occurring in chromatin (*Strom et al., 2017*; *Larson et al., 2017*; *Falk et al., 2019*; *Hilbert et al., 2021*). Moreover, a dynamic disordered bi-continuous phase separation is also in line with all the mentioned scenarios, especially considering that all imaging experiments are restricted to a quasi 2D slab of the system that could be insufficient to reveal a full 3D topology.

For the analysis of the SR-EV configurations, we take advantage of the methodology developed by our experimental collaborators and transform our coordinates to a stack of images (*Li et al., 2022*; *Li et al., 2021*). For this transformation, each bead is represented by a normal distribution and its contribution to a given voxel of the tomogram is the integral of the normal distribution over the voxel volume. We include *Figure 4—video 1* that is an example of the resulting volumetric image stack. As we display in *Figure 4B*, the image representation of the SR-EV conformations immediately reveals, in 2D, the inhomogeneity of the chromatin density that includes multiple regions of high density that we identify as packing domains. We analyzed the distribution of packing domain radii using the procedure outlined in *Figure 7—figure supplements 1 and 2*, which is essentially the same as the experimental one. In *Figure 7A*, we display the distribution of domain radii for all simulated conditions and the mean value for the 12 cases is displayed in *Figure 7B*. For comparison, we include in *Figure 7C* the results from our experiments on an A549 cell line (*Li et al., 2022*) obtained with ChromSTEM that

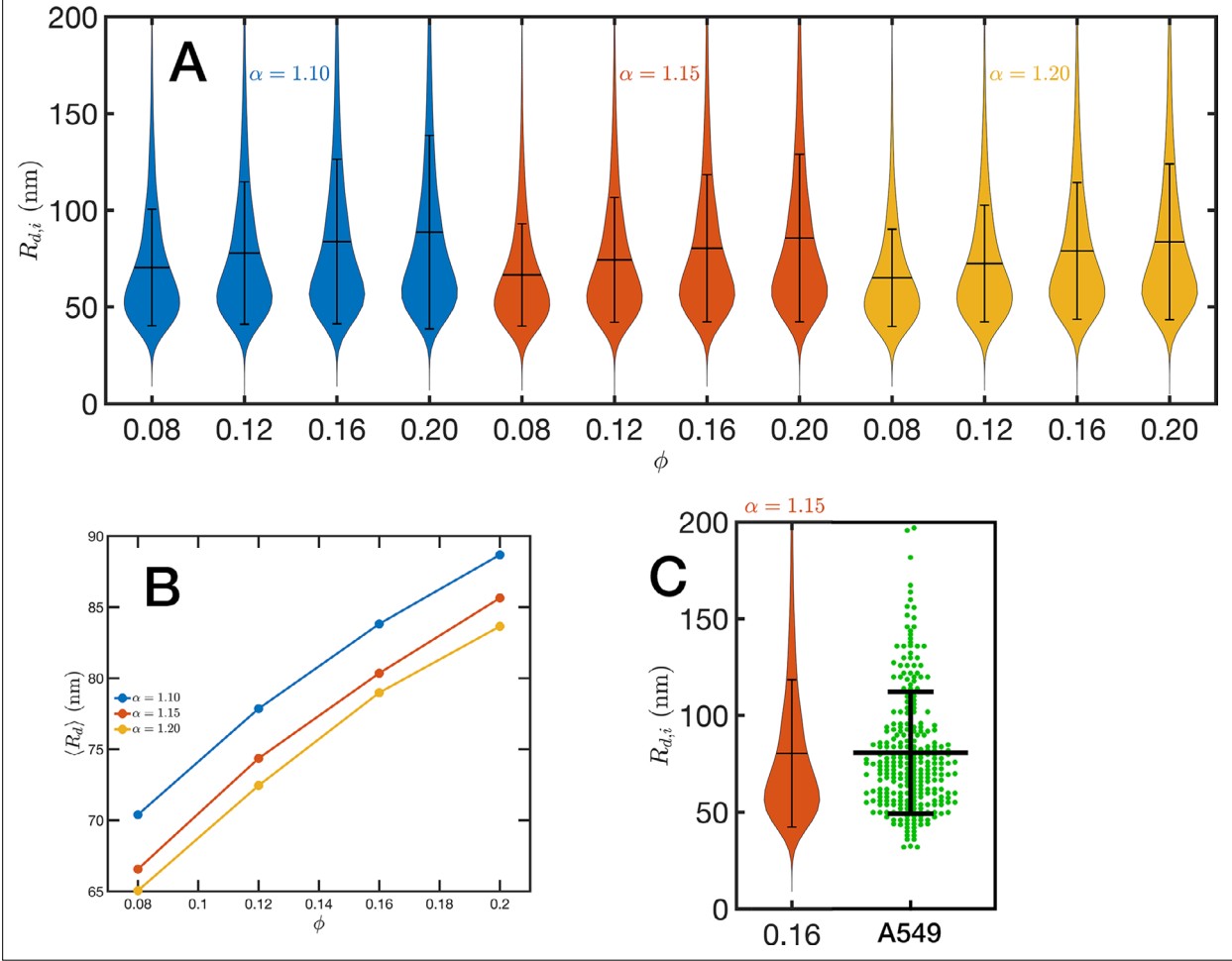

**Figure 7.** Chromatin packing domains. (**A**) Distributions of domain radii $R_{d,i}$ for all combinations of Self Returning Excluded Volume (SR-EV) parameters $\alpha$ and $\phi$, as labeled in the figure. (**B**) Mean value $\langle R_d \rangle$ of the domain radii distributions. (**C**) In green, experimental distribution of domain radii obtained with chromatin scanning transmission electron microscopy (ChromSTEM) on A549 cell line, and the closest approximation from SR-EV that corresponds to $\alpha = 1.15$ and $\phi = 0.16$.

The online version of this article includes the following figure supplement(s) for figure 7:

**Figure supplement 1.** Example of domain and domain's center determination from Self Returning Excluded Volume (SR-EV) slabs.

**Figure supplement 2.** Example of the determination of the density profiles of domains and their effective radii.

agree very well with the theoretical values in general, and in particular the agreement is excellent with the case corresponding to $\alpha = 1.15$ and $\phi = 0.16$.

In order to further characterize the structure of the model chromatin we calculated the pair correlation function between the model nucleosomes, i.e. $g(r)$. From the model definition and previous analysis, we know that $g(r)$ must reveal different features at different length scales. At short distances, $r \lesssim 40$ nm, $g(r)$ shows the structure of the dense packing domains through the typical maxima and minima, at the intermediate distances corresponding to the average size of the packing domains and the transition between intra- and inter-domains $g(r)$ is a decreasing function of $r$ approaching the expected plateau for large distances. Motivated by the mass scaling analysis introduced in Chrom-STEM experiments (**Li et al., 2022**; **Li et al., 2021**) we will use the integral form of the pair correlation function: $G(r) = \int_0^r 4\pi r'^2 g(r')dr'$. $G(r)$ smoothes out the short distance oscillations of $g(r)$ and reflects the intermediate regime as a power law with exponent $D < 3$.

In **Figure 8A**, we show in a log–log representation, as an example, the ensemble average $\langle G(r) \rangle$ corresponding to the global volume fraction $\phi = 0.16$ and the three values of $\alpha$. Between 40 and 120 nm we found that the $\langle G(r) \rangle$ is essentially a perfect straight line, i.e. $\langle G(r) \rangle \propto r^D$. We define $D$ as

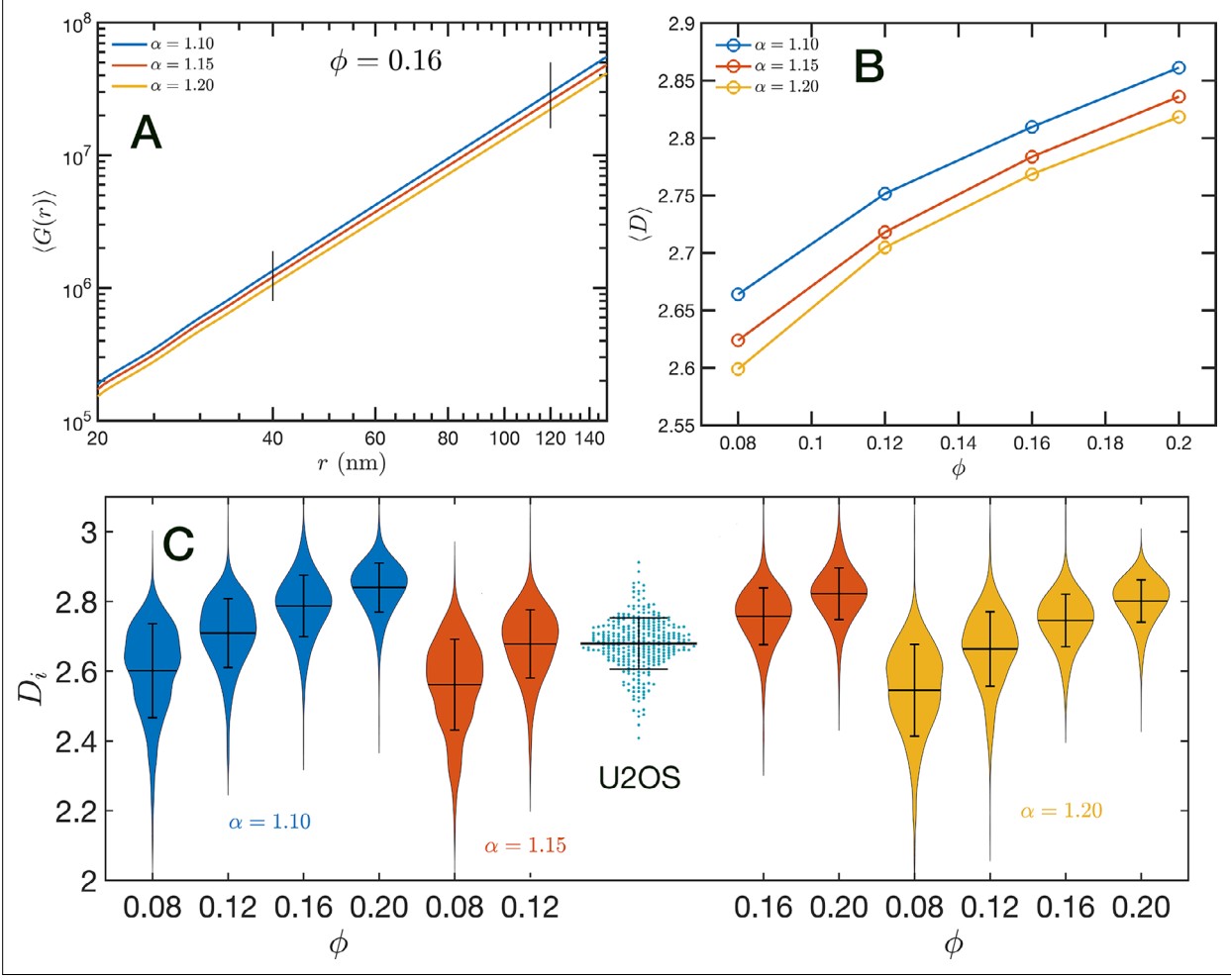

**Figure 8.** Packing coefficient D. (**A**) Ensemble average cumulative pair correlation function $\langle G(r) \rangle$ for $\phi = 0.16$ and the three studied values of $\alpha$. The vertical black lines mark the boundaries used to perform a power-law regression to calculate D. (**B**) Packing coefficient $\langle D \rangle$ as a function of $\phi$ and $\alpha$. (**C**) Distribution of packing coefficient $D_i$ for all the individual configurations for the 12 simulated conditions and, for comparison, we inserted the experimental partial wave spectroscopic (PWS) D results for U2OS cells that agree very well with the SR-EV results for $\phi = 0.12$ and $\alpha = 1.15$.

The online version of this article includes the following figure supplement(s) for figure 8:

**Figure supplement 1.** Example of cumulative distribution functions, $G_i(r)$, for five different Self Returning Excluded Volume (SR-EV) configurations.

the packing parameter that we calculate for $40 < r/\text{nm} < 120$. The slopes for the three displayed cases are slightly different, with D values ranging between 2.75 and 2.80 as $\alpha$ decreases from 1.20 to 1.10. In **Figure 8B**, we summarize the results for D for all the simulated conditions, which shows that D has a positive correlation with $\phi$, the overall volume fraction of the whole configuration, and a weaker inverse dependence on the folding parameter $\alpha$.

A similar power-law regression can be applied on the $G_i(r)$ obtained for each configuration. We use the subscript $i$ to distinguish that the quantity corresponds to a single configuration $i$. Since the configurations are obtained using a stochastic procedure, there is a large variability in the power-law fits obtained from them and some examples are included in **Figure 8—figure supplement 1**. In **Figure 8C**, we show the distributions of $D_i$ values for all 12 simulated conditions. Notice that individual $D_i$ can be larger than 3. To understand this in the context of population heterogeneity of chromatin structure, we performed live-cell PWS microscopy on U2OS cells and measured the distribution of chromatin packing states observed. As demonstrated from SR-EV, variations packing arise from the same $\alpha$ and $\phi$ conditions due to the probabilistic nature of the model. Consequently, this demonstrates that population heterogeneity arises intrinsically from our model, a finding consistent

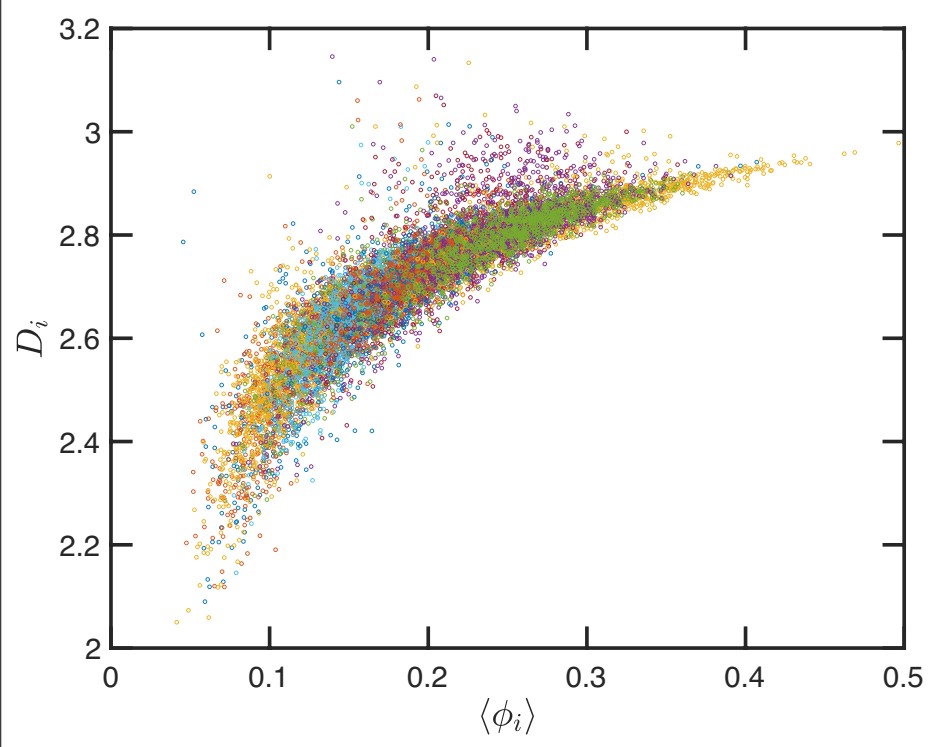

**Figure 9.** Local correlation between packing parameter and chromatin volume concentration. Relation between the calculated $D_i$ with the average local volume fraction $\langle\phi_i\rangle$. Both quantities are calculated for the same configuration and in the same spherical region of 240 nm in radius. The figure includes one point for each one of the 12,000 configurations of the 12 simulated ensembles.

with experimental results (*Figure 8C*). This heterogeneity arises in simulated conditions and is in best agreement for U2OS cells in the condition of $\alpha$ of 1.15 and $\phi$ of 0.16.

Up to this point, we have performed our analysis based on the SR-EV parameters $\alpha$ and to distinguish the different ensembles of configurations. However, the local volume fraction, as it has been shown in *Figures 2 and 6*, fluctuates at the scale of the packing domain size. This inhomogeneity makes the representation of a configuration by its overall SR-EV parameter $\phi$ not completely meaningful when we study a local or mesoscopic property, such as the packing parameter $D_i$. Therefore, it is convenient to introduce the local average chromatin volume fraction $\langle\phi_i\rangle$ calculated in exactly the same 240 nm sphere that we use to calculate $D_i$. The correlation between these two mesoscopic quantities is plotted in *Figure 9* and includes every one of the SR-EV 12,000 configurations. There is a very clear and interesting correlation between $D_i$ and $\langle\phi_i\rangle$. For high $\langle\phi_i\rangle$, the local $D_i$ approaches to 3, which is the theoretical upper limit for $\langle D\rangle$. For intermediate and small $\langle\phi_i\rangle$, there is a quite wide distribution of $D_i$ values, consistent with the violin plots of *Figure 8C*. Nevertheless, the local chromatin volume fraction is the main factor determining the corresponding packing parameter. In the next portion, we will demonstrate the predictions of SR-EV by affecting the probability of returns and the contribution of excluded volume by depleting RAD21 in HCT-116 cells.

## SR-EV predicts that loss of cohesin-mediated loops have a limited impact on packing domain formation

So far, we have presented the integration of $\alpha$ with excluded volume effects on representative, and distinct, methods to measure chromatin structure (Hi-C, live-cell PWS microscopy, and ChromSTEM). It is evident from SR-EV that these methods probe different features of genomic organization, which using SR-EV, could potentially be converged. To test the role of $\alpha$ in the regulation of genome structure, we target processes that govern stochastic returns. Recent work has demonstrated that cohesin-mediated loops are short lived, with an individual loop existing in that configuration ~6 % of the time (*Gabriele et al., 2022*). Likewise, the process of forming long-range returns is not exclusive to

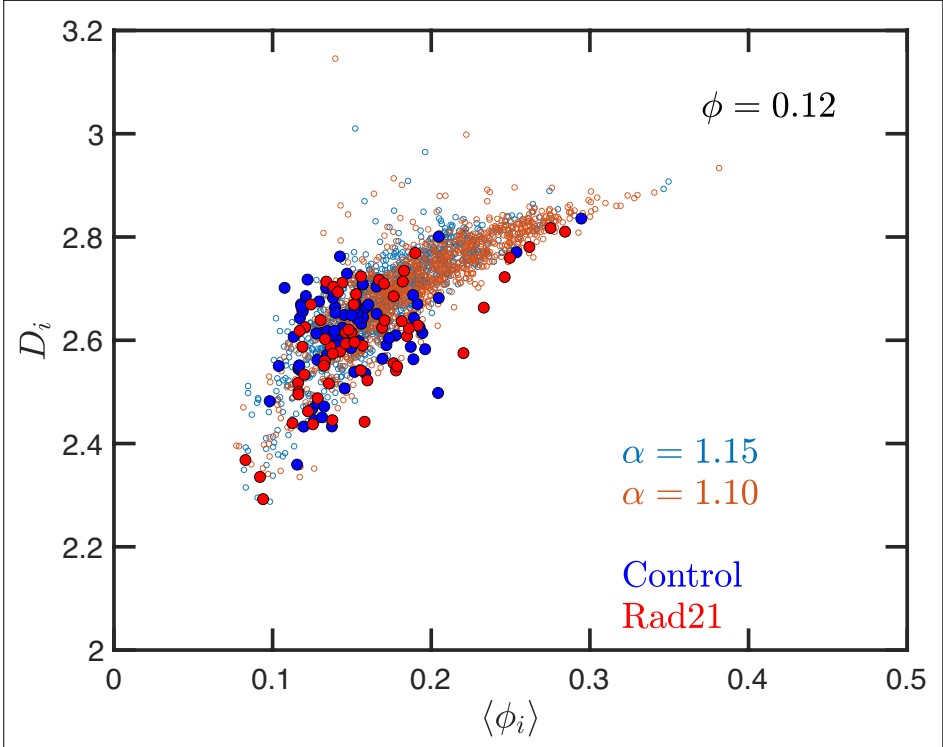

**Figure 10.** Effect of degrading RAD21 on the relation between packing parameter and chromatin volume concentration. The small open symbols are the Self Returning Excluded Volume (SR-EV) results for $\phi = 0.12$, $\alpha = 1.10$ and 1.15. The filled symbols represent the experimental values obtained with chromatin scanning transmission electron microscopy (ChromSTEM) (**Li, 2024**) for the control sample (blue) and the RAD21 degrade sample (red).

cohesin loop extrusion and arises from the confinement of a polymer in a crowded space as well as from transcription-induced promoter–promoter or promoter–enhancer interactions. Thus, even in the absence of cohesin or transcription, entropic loops would exist in chromatin. Consequently, all of these processes converge as components that produce a probability of a long-range step with a reciprocal probability of return at each individual loci across any individual cell. The returns and forward steps exist on a monomer scaffold and therefore individual nucleosome monomers cannot overlap in the same space.

Quantitively, SR-EV models the loss of cohesin-mediated loops as a decrease in $\alpha$ (e.g. from 1.20 to 1.15 or 1.10) without a change in nuclear volume at short-time scales ($\phi$ remains constant). SR-EV predicts (1) that domains will still exist on ChromSTEM (**Figure 7A**), (2) there will be a ~20% decrease in the number of domains, (3) the remaining domains will have similar sizes, densities, and mass scaling (**Figure 8C**), (4) population heterogeneity would be largely unaffected, and (5) $D$ in each domain will be predominantly determined by the local volume fractions (**Figures 8B and 9**). This view is in contrast to an alternative polymer model that produces biphasic structures from attractions coupled with loop extrusion (**Nuebler et al., 2018**), as the loss of cohesin in this model would results in microphases occurring only at large (~500 kbp) length scales. To test this experimentally, we degraded RAD21 (**Nishimura et al., 2009**; **Yesbolatova et al., 2020**) using HCT116-Rad21-mAID2 cell line and performed ChromSTEM and live-cell PWS imaging of these cells in comparison to a vehicle treated control at 4 hr. As predicted and in contrast to expectation from existing chromatin polymer models, we found that (see **Li, 2024**) the majority of chromatin packing domains are retained (~80% 62/78) with small changes in density (CVC 0.4 → 0.43), radius (84 → 89 nm), and (2.61 → 2.60). At the level of the heterogeneity of chromatin states observed in live cells, we performed live-cell PWS microscopy on cells with and without RAD21 depletion and find that it has a minimal impact on chromatin population diversity. Finally, we tested the prediction that local excluded volumes will predominantly determine the power-law geometry of chromatin within the nucleus. As observed on ChromSTEM, we find

that the local volume concentrations will non-linearly relate to the scaling behavior of the chromatin polymer while being minimally influenced by the change in $\alpha$ (**Figure 10**).

## Discussion

We have presented the SR-EV model based on stochastic rules of return and excluded volume interactions. Remarkably, the proposed rules of return are sufficient to generate polymer configurations having 3D packing domains that are observed in single-cell imaging experiments. We demonstrate that the SR-EV model produces chromosome-size configurations with nucleosome size monomers (200 bp) that agree with multiple distinct experimental methodologies spanning both live (PWS microscopy) and fixed cells (Hi-C, ChromSTEM) without the need for constraints from other -omic methodologies (e.g. ATAC-seq, ChIP-Seq). Without the need for these biological inputs, SR-EV configurations have contact $S < -1$, biphasic heterogeneous packing domains with a continuous distribution of sizes and densities, and the population heterogeneity innate to cellular systems. Initially, this seemed like a disquieting feature of the SR-EV model: stochastic returns based on a mathematical framework disregarding many long-held assumptions about hierarchical chromatin assembly produced strong experimental agreement, suggesting that genome organization is disordered at supranucleosome scales. This could create a paradox of how non-random features arise within organs (e.g. muscle is distinctly not the same as an eye) if one were to incorrectly equate stochasticity with randomness. Instead, we posit that stochastic returns are not synonymous solely with cohesin-mediated loop extrusion but are an agnostic event arising from multiple possible biochemical processes: be that short-range monomeric attractions, promoter–enhancer interactions, loop extrusion, etc. As such, stochastic returns will still occur in the event of the loss of any one of these mechanisms, thereby acting as a failsafe to maintain a degree of organizational integrity. That non-random tissues arise from a stochastic polymer would be viewed as an emergent, but not well understood, phenomenon from the interactions between stimuli/transcription factor signaling and disordered genome structure that requires further investigation (**Oberbeckmann et al., 2024**). This permits tissues to both have ensemble functions and the diversity of transcriptional states that are observed experimentally; microscopically identical cells (immune cells, muscle cells, bone cells, etc.) having a well-described distribution of transcriptional states in multiple organs.

An unexpected and testable prediction from the SR-EV model that is not present in existing polymer-modeling frameworks is that methods that measure connectivity (e.g. Hi-C) would be relatively insensitive to the effects of volume concentrations (a finding in line with on recent experimental results by Yiu and Dekker) whereas methods that measure chromatin density (ChromEM, super-resolution imaging) would show profound changes in chromatin response to density. This latter point is of particular significance in multiple clinical contexts. For instance, in cancer, variations in nuclear size and density are the oldest, most widely preserved hallmark of malignancy whose functional consequence remain poorly understood (**Hansemann, 1890**). In the context of SR-EV, these features result in significant structural heterogeneity within a cell population that could not be predicted by existing models due to their muted effects in methods that measure connectivity. Likewise, cells with low densities (neurons, oocytes, senescent cells) would contrast to cells with high densities (sperm, lymphocytes) in their domain structure but would have relatively similar contact scaling behaviors. SR-EV would therefore predict domain organization arises from chromatin concentration and nuclear size directly, indicating that a global feature (nuclear size, CVC) would have mechanistic consequences that would otherwise be missed in the existing polymer framework of genome organization.

We present a novel model of chromatin based on stochastic returns and physical interactions that captures the ground-truth structures observed across both imaging and sequencing based measures of chromatin organization. By maintaining in SR-EV the possibility of self-returning extensions that are presented in SRRW, several features arise. (1) High frequency, short return events lead to the formation of individual packing domains. (2) Low frequency, large steps give rise to a corrugated chromatin structure at intermediate length scales (~100 nm) that allows genomic accessibility to arise (**Figure 2**). Expanding on the theory originally presented by **Huang et al., 2020**, we now can account for excluded volume interactions between single nucleosomes to quantitively and qualitatively represent chromatin configurations. This extension is crucial as it allows for the accurate reconstruction of the occupied volumes within chromatin and to calculate the physical properties of genomic organization. Pairing the excluded volume representation of the individual monomer units (nucleosomes) with

stochastic returns produces a continuous heterogeneous polymer chain with a random distribution of space-filling domains. In comparing the effects of the folding parameter, $\alpha$, with the overall chromatin volume fraction, $\phi$, we show that just two parameters can recapture the heterogeneous nature of chromatin observed in electron microscopy, the variations in CVCs, the formation of packing domains with appropriate sizes, that power-law distributions are present at intermediate length scales (quantified by $D$), and the heterogeneity observed experimentally in live-cell measurements of chromatin structure.

Crucially, the SR-EV model is grounded in the stochastic description of genome organization which allows capturing both the description of ensemble properties (e.g. populations of cells/chromosomes) and individual chromosomes. This feature is what allows both the accurate representation of individual experiments (such as the visualized 3D structure in ChromSTEM) as well as features that only become apparent over numerous realizations (such as contact scaling observed in Hi-C, population heterogeneity observed in PWS microscopy). The model unit length coincides with the size of a nucleosome and owing to physical principles, the linker unit produced is concordant with reported experimental values of 35–45 bp (*Table 1*). The present length of the model polymer is comparable with the size of human chromosome 16 or smaller; but could be expanded with additional computational resources. Therefore, the SR-EV configurations span over a large range of spatial dimensions (~10 nm to ~1 μm). The agreement with the experimentally found CVC distributions gives us a first confirmation on the validity of the model, and an indication of the relevant values for $\alpha$ and $\phi$ present physiologically. The quantitative agreement of the packing domain radii distribution with the outcome of ChromSTEM reinforce the confidence in the theory. The packing parameter $D$ is defined in terms of the incremental pair correlation function between model nucleosomes; a definition that is similar (but not exactly the same) as the one proposed in ChromSTEM studies. The value of $D$ is consistently found between 2 and 3 for all simulated conditions. $D$ is calculated on a mesoscopic region of 240 nm in radius, which is completely independent of the location of the packing domains. However, since we show that there is a strong positive correlation between $D_i$ and the corresponding local volume fraction $\langle \phi_i \rangle$ we can infer that regions containing large packing domains will be associated with a large $D$. The distribution of $D_i$ values span over the same range of values observed in PWS experiments. In particular, we show a case in excellent quantitative agreement with PWS results for U2OS cell line (noting that similar distributions are observed independently of this cancer cell line). The incorporation of genomic character to the SR-EV model will allow us to study all individual single chromosomes properties, and also topological associated domains and A/B compartmentalization from ensemble of configurations as in Hi-C experiments.

Finally, we view the simplicity of our model as a core strength as it already captures key details about genome organization without introducing many of the constraints present within existing frameworks. Currently, we could generate 12,000 independent configurations of a 500,000 nucleosome (75 Mbp, approximately the size of chromosome 16) within a short period of time. Likewise, we envision that future work can incorporate some of the myriad molecular features known to exist within chromatin organization to be able to interrogate how key components (e.g. sparse, focal constraint such as CTCF-binding sites or heterochromatin modifying enzymes) would alter the observed physical structures. As with any modeling work, there will always be the tension between the addition of details for fidelity and the ability to capture the properties of genome organization. As the SR-EV already captures many key properties seen within chromatin, we anticipate that it can serve as the basis model of stochastically configured genome organization within the wider field.

## Materials and methods
### Cell culture

Human cell line U2OS cells (ATCC, #HTB-96) used for experimental validation of the model were cultured in McCoy's 5A Modified Medium (Thermo Fisher Scientific, #16600-082) supplemented with 10% fetal bovine serum (FBS) (Thermo Fisher Scientific, #16000-044) and 100 μg/ml penicillin–streptomycin antibiotics (Thermo Fisher Scientific, #15140-122). Human cell line A549 cells (ATCC, #CCL-185) used for experimental validation of the model were cultured in Dulbecco's modified Eagle's medium (Thermo Fisher Scientific, #11965092) supplemented with 10% FBS (Thermo Fisher Scientific, #16000-044) and 100 μg/ml penicillin–streptomycin antibiotics (Thermo Fisher Scientific, #15140-122).

Experiments were performed on cells from passages 5 to 10. All cells were maintained under recommended conditions at 37°C and 5% $CO_2$. Cells were verified to have no detectable mycoplasma contamination (ATCC, #30-1012K) prior to starting experiments.

## PWS sample preparation

Prior to imaging, cells were cultured in 35 mm glass-bottom Petri dishes. All cells were allowed a minimum of 24 hr to re-adhere and recover from trypsin-induced detachment. PWS imaging was performed when the surface confluence of the dish was approximately 70%.

## PWS imaging

The PWS optical instrument consists of a commercial inverted microscope (Leica, DMIRB) equipped with a broad-spectrum white light LED source (Xcite-120 light-emitting diode lamp, Excelitas), ×63 oil immersion objective (Leica HCX PL APO, NA1.4 or 0.6), long pass filter (Semrock, BLP01-405R-25), and Hamamatsu Image-EM CCD camera C9100-13 coupled to an LCTF (CRi VariSpec). Live cells were imaged and maintained under physiological conditions (37°C and 5% $CO_2$) using a stage top incubator (In Vivo Scientific, Stage Top Systems). Briefly, PWS directly measures the variations in spectral light interference that results from internal light scattering within the cell, due to heterogeneities in chromatin density, with sensitivity to length scales between 20 and 300 nm (*Li et al., 2021*). Variations in the refractive index distribution are characterized by the mass scaling (chromatin packing scaling) parameter, $D$. A detailed description of these methods is reported in several publications (*Subramanian et al., 2009*; *Almassalha et al., 2016*; *Gladstein et al., 2018*; *Eid et al., 2020*).

## ChromSTEM sample preparation and imaging

Cell samples were prepared as reported in *Li et al., 2022*. Cells were first washed with Hank's Balanced Salt Solution without calcium and magnesium (Thermo Fisher Scientific, #14170112) three times, 2 min each. Fixation, blocking, DNA staining and 3,3'-diaminobenzidine tetrahydrochloride (DAB) solutions were prepared with 0.1 M sodium cacodylate buffer (pH = 7.4). Cells were fixed with 2% paraformaldehyde, 2.5% glutaraldehyde, 2 mM calcium chloride for 5 min in room temperature and 1 hr on ice and all the following steps were performed on ice or in cold temperature unless otherwise specified. After fixation, cells were washed with 0.1 M sodium cacodylate buffer five times, 2 min each. Cells were then blocked with 10 mM glycine, 10 mM potassium cyanide for 15 min. Cells were stained with 10 µM DRAQ5, 0.1% Saponin for 10 min and washed with the blocking solution three times 5 min each. Cells were bathed in 2.5 mM DAB and exposed to 150 W Xenon Lamp with ×100 objective lens and a Cy5 filter for 7 min. Cells were washed with 0.1 M sodium cacodylate buffer five times, 2 min each, followed by staining with 2% osmium tetroxide, 1.5% potassium ferrocyanide, 2 mM calcium chloride, 0.15 M sodium cacodylate buffer for 30 min. After osmium staining, cells were washed with double distilled water five times, 2 min each and sequentially dehydrated with 30%, 50%, 70%, 85%, 95%, 100% twice, ethanol, 2 min each. Cells were then washed with 100% ethanol for 2 min and infiltrated with Durcupan ACM ethanol solutions (1:1 for 20 min, and 2:1 for 2 hr) at room temperature. Cells were then infiltrated with resin mixture for 1 hr, resin mixture with accelerator for 1 hr in 50°C dry oven and embedded in BEEM capsule with the resin mixture at 60°C dry oven for 48 hr.

Resin sections with thickness around 100 nm were prepared with a Leica UC7 ultramicrotome and a 35°C DiATOME diamond knife. The sections were collected on copper slot grids with carbon/Formvar film and 10 nm colloidal gold nanoparticles were deposited on both sides of the section as fiducial markers. HAADF(High-angle annular dark-field imaging) images collected by a 200-kV cFEG Hitachi HD2300 scanning transmission electron microscope. For each sample, projections were collected from −60 to +60°C with 2°C increments, along two roughly perpendicular axes.

Each projection series along one rotation axis was aligned with IMOD using gold nanoparticle fiducial markers. After image alignment, penalized maximum likelihood algorithm in Tomopy was used to reconstruct the images with 40 iterations. IMOD was used to combine tomograms from different rotation axis of the same sample.

## Chromatin domain radius measured from experiment

The chromatin domains were identified using FIJI. 2D chromatin density distributions were obtained by re-projection of the tomogram along z-axis, followed by Gaussian filtering with 5 pixels radius and

CLAHE contrast enhancements with block size of 120 pixels. Chromatin domain centers were selected as the local maxima of chromatin density.

To evaluate the size of a domain, two properties were analyzed for each domain, which are the mass scaling properties and radial volume chromatin concentration (CVC). For mass scaling, multiple mass scaling curves were sampled by using pixels (a 11-pixel × 11-pixel window) around the center of an identified domain and they were averaged by the weight of the pixel values of the selected center pixel. A size of domain is defined by the length scale that the domain meets any of the following three criteria: (1) it deviates from the power-law mass scaling relationship $M(r) \propto r^D$ by 5%; (2) the local fitting of $D$ reaches 3; (3) the radial CVC reaches a local minimum and begins to increase for longer length scale.

### Experimental validation plots

GraphPad Prism 10.0.0 was used to make the violin plots in *Figures 7C and 8C*. The violin plots are represented as individual data points, with lines at the median and quartiles.

## Acknowledgements

We acknowledge funding from the National Institutes of Health (NIH) grants U54CA268084, U54CA261694, R01CA228272, R01CA224911, R01CA225002, T32GM142604, NSF grant EFMA-1830961, and philanthropic support from K Hudson and R Goldman, S Brice and J Esteve, ME Holliday and I Schneider, the Christina Carinato Charitable Foundation, and D Sachs. Luay Almassalha acknowledges the support of NIH training grant T32AI083216. This research was supported in part through the computational resources and staff contributions provided for the Quest high performance computing facility at Northwestern University which is jointly supported by the Office of the Provost, the Office for Research, and Northwestern University Information Technology.

## Additional information

### Funding

| Funder | Grant reference number | Author |
| --- | --- | --- |
| National Cancer Institute | U54CA268084 | Vadim Backman<br>Igal Szleifer |
| National Cancer Institute | U54CA261694 | Vadim Backman |
| National Cancer Institute | R01CA228272 | Vadim Backman |
| National Cancer Institute | R01CA224911 | Vadim Backman |
| National Cancer Institute | R01CA225002 | Vadim Backman |
| National Institute of General Medical Sciences | T32GM142604 | Vadim Backman |
| National Science Foundation | EFMA-1830961 | Vadim Backman<br>Igal Szleifer |
| National Institute of Allergy and Infectious Diseases | T32AI083216 | Luay M Almassalha |

The funders had no role in study design, data collection, and interpretation, or the decision to submit the work for publication.

### Author contributions

Marcelo A Carignano, Conceptualization, Data curation, Software, Formal analysis, Investigation, Visualization, Methodology, Writing – original draft, Writing – review and editing; Martin Kroeger, Conceptualization, Data curation, Software, Investigation, Visualization, Methodology, Writing – original draft, Writing – review and editing; Luay M Almassalha, Conceptualization, Formal analysis, Methodology, Writing – original draft, Writing – review and editing; Vasundhara Agrawal, Wing Shun Li, Emily M Pujadas-Liwag, Validation; Rikkert J Nap, Validation, Investigation; Vadim Backman,

Conceptualization, Supervision, Funding acquisition, Writing – original draft, Writing – review and editing; Igal Szleifer, Conceptualization, Supervision, Funding acquisition, Methodology, Writing – original draft, Writing – review and editing

Author ORCIDs
Marcelo A Carignano ⓘ https://orcid.org/0000-0001-8345-7724
Luay M Almassalha ⓘ https://orcid.org/0000-0001-9355-7681
Vasundhara Agrawal ⓘ https://orcid.org/0000-0003-0913-9298
Vadim Backman ⓘ https://orcid.org/0000-0003-1981-1818
Igal Szleifer ⓘ https://orcid.org/0000-0002-8708-0335

Reviewer #1 (Public review): https://doi.org/10.7554/eLife.97604.3.sa1
Author response https://doi.org/10.7554/eLife.97604.3.sa2

Data availability
The current manuscript is a computational study based on our own software.

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

## Appendix 1

### Supplementary algorithms

#### Algorithm for the generation of an SRRW in free space

Here, we describe a recursive Monte Carlo algorithm to generate an SRRW parameterized by folding parameter $\alpha > 1$ and local cutoff $U_{\max}$, which represents the maximum bond length. Within the algorithms to be described, the length unit is the minimum bond length $U_{\min} = 10$ nm, so that all length are dimensionless, and taken relative to the minimum bond length $U_{\min}$. The conformation of the SRRW, emanating from the origin, is defined by $N$ bond vectors $\mathbf{U}_1$, $\mathbf{U}_2$, …, $\mathbf{U}_N$. In the following, the symbol $\xi$ stands for an independent random number drawn with equal probability from the interval $[0, 1]$, and has to be recreated whenever it occurs below.

(A1) Define $\beta \equiv 1 - U_{\max}^{-(1+\alpha)}$.
(A2) Generate a set of $2N$ bond vectors $\mathbf{B}_n$ with $n = 1, 2, .., 2N$ for eventual later use. Each $\mathbf{B}_n$ is given by $\mathbf{B}_n = \ell\mathbf{u}$, where $\mathbf{u}$ is a random unit vector and $\ell = (1 - \beta\xi)^{-1/(1+\alpha)}$ its bond length. A random unit vector, we create via $\mathbf{u} = (\sqrt{1 - z^2} \cos\phi, \sqrt{1 - z^2} \sin\phi, z)$, where $\phi = 2\pi\xi$ and $z = 2\xi - 1$. The generation of the set $\{\mathbf{B}\}$ hence requires $6N$ random numbers $\xi$ and if not otherwise mentioned, the $\{\mathbf{B}\}$ will remain unchanged during the course of the algorithm.
(A3) Initialize $n = N$, set $\mathbf{U}_1 = \mathbf{B}_n$.
(A4) Increase $n$ by one, set $\mathbf{U}_2 = \mathbf{B}_n$, and initialize step $s = 2$.
(A5) Call a recursive routine that takes the existing sets $\{\mathbf{B}\}$, $\{\mathbf{U}\}$, $n$, and $s$ as arguments, and returns new sets $\{\mathbf{B}\}$, $\{\mathbf{U}\}$, and $n$. This routine does the following:

If $s = N$, just return from the routine.
Calculate return probability $P_R = |\mathbf{U}_s|^{-\alpha}/\alpha$.
If $\xi < P_R$, then $\mathbf{U}_{s+1} = -\mathbf{B}_n$ and $n$ is decreased by one. Otherwise, $n$ is increased by one, the single $\mathbf{B}_n$ is re-created using the above procedure (A2), and $\mathbf{U}_{s+1} = \mathbf{B}_n$.
Routine calls itself with identical arguments as before, with the exception of $s + 1$ instead of $s$.

The described algorithm terminates automatically as soon as $N$ bond vectors $\mathbf{U}_1$, $\mathbf{U}_2$, …, $\mathbf{U}_N$ have been created. The coordinates $\{\mathbf{x}\}$ of nodes are simply given by the cumulative sum over the set of bond vectors $\{\mathbf{U}\}$, i.e. $\mathbf{x}_{j+1} = \mathbf{x}_j + \mathbf{U}_j$. Note that using this algorithm the return probabilities satisfy *Equation 1* and that all bond lengths $\ell$ are automatically confined to the interval $[U_{\min}, U_{\max}]$ and distributed according to *Equation 2*. The proof is provided in the next section.

#### Algorithm for the generation of an SRRW subject to global cutoff

The idea of an SRRW with global cutoff $R_c$ is to make sure the SRRW will tend to grow within a certain spherical volume of radius $\approx R_c$. To this end the above algorithm is slightly modified as follows. Instead of the earlier (ii) calculate the geometric center $\mathbf{C}$ of the existing nodes from $\{\mathbf{x}\}$. If $|\mathbf{x}_s - \mathbf{C}| > R_c$, then set, otherwise calculate $P_R = |\mathbf{U}_s|^{-\alpha}/\alpha$ as before.

#### Molecular dynamics protocol for the generation of an SR-EV

An SRRW conformation subject to global cutoff is produced via Monte Carlo as just described; such a conformation usually exhibits a large number of nodes (points) with identical coordinates. All these points need to be turned into beads, i.e., receive a finite spherical volume within the final SR-EV configuration, that should preserve all large scale features and domain characteristics of the SRRW. We alter the local structure to avoid bead–bead overlap, while operating at (ideally) minimal displacement effort. To this end we use the original node coordinates $\{\mathbf{x}\}$ as initial center positions of spherical beads of radius $r_\circ = 0.49$ and unit mass $m$. In a first step, to allow for a random element, and to avoid center–center distances that are exactly zero up to numerical precision, we displace all overlapping beads randomly by 1% of the bead diameter. Afterwards we employ LAMMPS (*Thompson et al., 2022*) to run a molecular dynamics simulation on the modified SRRW systems composed of spherical beads. We let all beads interact via a soft repulsive radially symmetric pair potential $V(r) = 20\epsilon[1 + \cos(\pi r/r_c)]$ for $r \leq r_c$, and $V(r) = 0$ otherwise, where $r$ denotes the center–center distance between pairs of beads, $\epsilon$ the irrelevant energy unit, and the cutoff distance $r_c = 1.03$ is chosen slightly larger than the bead diameter. The system is thermostatted via the Nosé–Hoover

scheme at $T = 0.001\,\epsilon/k_B$, and run using a time step $\Delta t = 0.005\,U_{min}\sqrt{m/\epsilon}$. During runtime, the bead–bead pair correlation function $g(r)$ is evaluated at each time step and averaged for a duration of 200 time steps. Each time unit (200 time steps) we inspect the averaged $g(r)$, integrated up to $r_c$, as this quantity informs about the amount of remaining overlap. In rare cases, the integral did not decrease with time, in that case we start over using another seed value for the random number generator. While the integral keeps decreasing, we monitor the potential energy of the system. As soon as the potential energy has reached a minimum, which happens if the energy is close to zero, we terminate the molecular dynamics run and save the resulting SR-EV coordinates. The minimum center–center distance between pairs of beads in the SR-EV configuration exceeds $2r_o$, as we verified. Note that the distribution of bond lengths is significantly different for SR-EV and SRRW conformations.

## Proof of the validity of the SRRW algorithm

The forward jump probability $P_J(U) = (\alpha + 1)U^{-(\alpha+2)}$ was stated in the manuscript. It was furthermore mentioned that new bonds of length $U_1$ should not exceed a local dimensionless cutoff length $U_{max}$, while $U_{min} = 1$ within these units. Because $P_J$ is a probability distribution, it must fulfill $\int_1^{U_{max}} P_J(U)dU = 1$ and the properly normalized version thus reads

$$P_J(U) = \frac{(\alpha + 1)U^{-(\alpha+2)}}{1 - U_{max}^{-(\alpha+1)}}, \qquad U \in [1, U_{max}]. \tag{3}$$

To efficiently create bond lengths $U$ distributed according to *Equation 3* using equally distributed random numbers $\xi \in [0, 1]$, one has to solve the differential equation $\xi'(U) = P_J(U)$ with initial condition $\xi(1) = 0$, and then invert the solution. The solution of the differential equation is $\xi(U) = (U_{max}/U)^{1+\alpha}(U^{1+\alpha} - 1)/(U_{max}^{1+\alpha} - 1)$. Solving this expression for $U$ gives $U = (1 - \beta\xi)^{-1/(1+\alpha)}$ with the constant $\beta \equiv 1 - U_{max}^{-(1+\alpha)}$, so that $U = 1$ and $U = U_{max}$ for $\xi = 0$ and $\xi = 1$, respectively. This completes the proof of item (A2) with (A1).

It might be just interesting to mention that one has access to some statistical properties of the chain conformation from $\beta \equiv 1 - U_{max}^{-(1+\alpha)}$, while $P_R$ has to be taken into account for the exact calculation. For sufficiently large $U_{max}$ the mean bond length is

$$\langle U \rangle = \int_1^{U_{max}} U P_J(U)dU \approx \frac{1+\alpha}{\alpha}. \tag{4}$$

For $\alpha \in \{1.1, 1.15, 1.2\}$ the mean bond length is hence $\langle U \rangle \in \{1.91, 1.87, 1.83\}$. Similarly, the mean return probability is approximately

$$\langle P_R \rangle = \int_1^{U_{max}} P_R(U) P_J(U)\, dU \approx \frac{1+\alpha}{\alpha(1+2\alpha)}, \tag{5}$$

i.e. $\langle P_R \rangle = \{0.597, 0.567, 0.539\}$ for $\alpha = \{1.1, 1.15, 1.2\}$. While for $\alpha \leq 1.03$ the SRRW basically collapses to a small region in space, beyond this value the effective number of forward steps is approximately $[0.49(\alpha - 1) - 0.02]N \approx (\alpha - 1)N/2$.

