## [Editor Report · eLife assessment]

The authors develop a self-returning self-avoiding polymer model of chromosome organization and show that their framework can recapitulate at the same time local density and large-scale contact structural properties observed experimentally by various technologies. The presented theoretical framework and the results are **valuable** for the community of modelers working on 3D genomics. The work provides **solid** evidence that such a framework can be used, is reliable in describing chromatin organization at multiple scales, and could represent an interesting alternative to standard molecular dynamics simulations of chromatin polymer models.

---

## [Referee Report · Reviewer #1 (Public review)]

Carignano et al propose an extension of the self-returning random walk (SRRW) model for chromatin to include excluded volume aspects and use it to investigate generic local and global properties of the chromosome 3D organization inside eukaryotic nuclei. In particular, they focus on chromatin volumic density, contact probability and domain size and suggest that their framework can recapitulate several experimental observations and predict the effect of some perturbations.

Strengths:

• The developed methodology is convincing and may offer an alternative - less computationally demanding - framework to investigate the single-cell and population structural properties of 3D genome organization at multiple scales.

• Compared to the previous SRRW model, it allows for investigation of the role of excluded volume locally.

• They perform some experiments to compare with model predictions and show consistency between the two.

Weaknesses:

• The model currently cannot fully account for specific mechanisms that may shape the heterogeneous, complex organization of chromosomes (TAD at specific positions, A/B compartmentalization, promoter-enhancer loops, etc.).

• By construction of their framework, excluded volume only impacts locally the polymer organization and larger-scale properties for which excluded volume could be a main actor (formation of chromosome territories [Rosa & Everaers, PLoS CB 2009], bottle-brush effects due to loop extrusion [Polovnikov et al, PRX 2023], etc.) cannot be captured.

• Comparisons with experiments are solid but are not clearly quantified.

Impact:

Building on the presented framework in the future to incorporate TAD and compartments may offer an interesting model to study the single-cell heterogeneity of chromatin organization. But currently, in this reviewer's opinion, standard polymer modeling frameworks may offer more possibilities.

---

## [Author Response]

The following is the authors’ response to the original reviews.

**eLife assessment**
The authors develop a self-returning self-avoiding polymer model of chromosome organization and show that their framework can recapitulate at the same time local density and large-scale contact structural properties observed experimentally by various technologies. The presented theoretical framework and the results are valuable for the community of modelers working on 3D genomics. The work provides solid evidence that such a framework can be used, is reliable in describing chromatin organization at multiple scales, and could represent an interesting alternative to standard molecular dynamics simulations of chromatin polymer models.

We appreciate the editor for an accurate description of the scope of the paper.

**Public Reviews:**

**Reviewer #1 (Public Review):**
Carignano et al propose an extension of the self-returning random walk (SRRW) model for chromatin to include excluded volume aspects and use it to investigate generic local and global properties of the chromosome 3D organization inside eukaryotic nuclei. In particular, they focus on chromatin volumic density, contact probability, and domain size and suggest that their framework can recapitulate several experimental observations and predict the effect of some perturbations.

We thanks the reviewer for the attention paid to the manuscript and all the relevant comments.

Strengths:- The developed methodology is convincing and may offer an alternative - less computationally demanding - framework to investigate the single-cell and population structural properties of 3D genome organization at multiple scales.- Compared to the previous SRRW model, it allows for investigation of the role of excluded volume locally.

Excluded volume is accounted for everywhere, not locally. We emphasized this on page 3, line 182:

“The method that we employ to remove overlaps is a low-temperature-controlled molecular dynamics simulation using a soft repulsive interaction potential between initially overlapping beads, that is terminated as soon as all overlaps have been resolved, as described in the Appendix 3.”

- They perform some experiments to compare with model predictions and show consistency between the two.Weaknesses:- The model is a homopolymer model and currently cannot fully account for specific mechanisms that may shape the heterogeneous, complex organization of chromosomes (TAD at specific positions, A/B compartmentalization, promoter-enhancer loops, etc.).

The SR-EV model is definitely not a homo-polymer, as it is not a regular concatenation of a single monomeric unit.

The model includes loops, which may happen in two ways: (1) As in the SRRW, branching structures emerging from the configuration backbone can be interpreted as nested loops and (2) A relatively long forward step followed by a return is a single loop. The model induces the formation of packing domains, which are not TADs, and are quantitatively in agreement with ChromSTEM experiments.

We consider convenient to add a new figure that will further clarify the structures obtained with the SR-EV model. The following paragraph and figure has been added in page 5:

“The density heterogeneity displayed by the SR-EV configurations can be analyzed in terms of the accessibility. One way to reveal this accessibility is by calculating the coordinations number (CN) for each nucleosome, using a coordination radius of 11.5 nm, along the SR-EV configuration. CN values range from 0 for an isolated nucleosome to 12 for a nucleosome immersed in a packing domain. In Figure 3 we show the SR-EV configuration showed in Figure 2, but colored according to CN. CN can be also considered as a measure to discriminate heterochromatin (red) and euchromatin (blue). Figure 3-A shows how the density inhomogeneity is coupled to different CN, with high CN represented in red and low CN represented in blue. Figure 3-B show a 50 nm thick slab obtained from the same configuration that clearly show the nucleosomes at the center of each packing domains are almost completely inaccesible, while those outside are open and accessible. It is also clear that the surface of the packing domains are characterized by nearly white nucleosomes, i.e. coordinated towards the center of the domain and open in the opposite direction.”

- By construction of their framework, the effect of excluded volume is only local and larger-scale properties for which excluded volume could be a main actor (formation of chromosome territories [Rosa & Everaers, PLoS CB 2009], bottle-brush effects due to loop extrusion [Polovnikov et al, PRX 2023], etc.) cannot be captured.

Excluded volume is considered for all nucleosomes, including overlapping beads distant along the polymer chain. Chromosome territories can be treated, but it is not in this case because we look at a single model chromosome.

- Apart from being a computationally interesting approach to generating realistic 3D chromosome organization, the method offers fewer possibilities than standard polymer models (eg, MD simulations) of chromatin (no dynamics, no specific mechanisms, etc.) with likely the same predictive power under the same hypotheses. In particular, authors often claim the superiority of their approach to describing the local chromatin compaction compared to previous polymer models without showing it or citing any relevant references that would show it.

We apologize if the text transmit an idea of superiority over other methods that was not intended. SR-EV is an alternative tool that may give a different, even complementary point of view, to standard polymer models.

- Comparisons with experiments are solid but are not quantified.

The comparisons that we have presented are quantitative. We do not have so far a way to characterize alpha or phi, a priori, for a particular system.

Impact:Building on the presented framework in the future to incorporate TAD and compartments may offer an interesting model to study the single-cell heterogeneity of chromatin organization. But currently, in this reviewer's opinion, standard polymer modeling frameworks may offer more possibilities.

We thank the reviewer for the positive opinion on the potential of the presented method. The incorporation of TADs and compartments is left for a future evolution of the model as its complexity will make this work extremely long.

**Reviewer #2 (Public Review):**
Summary:The authors introduce a simple Self Returning Excluded Volume (SR-EV) model to investigate the 3D organization of chromatin. This is a random walk with a probability to self-return accounting for the excluded volume effects. The authors use this method to study the statistical properties of chromatin organization in 3D. They compute contact probabilities, 3D distances, and packing properties of chromatin and compare them with a set of experimental data.

We thank the reviewer for the attention paid to our manuscript.

Strengths:(1) Typically, to generate a polymer with excluded volume interactions, one needs to run long simulations with computationally expensive repulsive potentials like the WeeksChanlder-Anderson potential. However, here, instead of performing long simulations, the authors have devised a method where they can grow polymer, enabling quick generation of configurations.(2) Authors show that the chromatin configurations generated from their models do satisfy many of the experimentally known statistical properties of chromatin. Contact probability scalings and packing properties are comparable with Chromatin Scanning Transmission Electron Microscopy (ChromSTEM) experimental data from some of the cell types.Weaknesses:This can only generate broad statistical distributions. This method cannot generate sequence-dependent effects, specific TAD structures, or compartments without a prior model for the folding parameter alpha. It cannot generate a 3D distance between specific sets of genes. This is an interesting soft-matter physics study. However, the output is only as good as the alpha value one provides as input.

We proposed a model to create realistic chromatin configuration that we have contrasted with specific single cell experiments, and also reproducing ensemble average properties. 3D distances between genes can be calculated after mapping the genome to the SR-EV configuration. The future incorporation of the genome sequence will also allow us to describe TADs and A/B compartments. See added paragraph in the Discussion section:

“The incorporation of genomic character to the SR-EV model will allow us to study all individual single chromosomes properties, and also topological associated domains and A/B compartmentalization from ensemble of configurations as in HiC experiments. “

**Recommendations for the authors:**

**Reviewer #1 (Recommendations For The Authors):**
Major:- In the introduction and along the text, the authors are often making strong criticisms of previous works (mostly polymer simulation-based) to emphasize the need for an alternative approach or to emphasize the outcomes of their model. Most of these statements (see below) are incomplete if not wrong. I would suggest tuning down or completely removing them unless they are explicitly demonstrated (eg, by explicit quantitative comparisons). There is no need to claim any - fake - superiority over other approaches to demonstrate the usefulness of an approach. Complementarity or redundance in the approaches could also be beneficial.

We regret if we unintentionally transmitted a claim of superiority. We have made several small edits to change that.

- Line 42-43: at least there exist many works towards that direction (including polymer modeling, but also statistical modeling). For eg, see the recent review of Franck Alber.

Line removed. Citation to Franck Alber included below in the text.

- Line 54-57: Point 1 is correct but is it a fair limitation? These models can predict TADs & compartments while SR-EV no. Point 2 is wrong, it depends on the resolution of the model and computer capacity but it is not an intrinsic limitation. Point 3 is wrong, such models can predict very well single-cell properties, and again it is not an intrinsic limitation of the model. Point 4 is incorrect. The space-filling/fractal organization was an (unfortunate) picture to emphasize the typical organization of chromosomes in the early times (2009), but crumpled polymers which are a more realistic description are not space-filling (see Halverson et al, 2013).

Text involving points 1 to 4 removed. It was unnecessary and does not change the line of the paper.

- L400-402 + 409-411: in such a model, the biphasic structure may emerge from loop extrusion but also naturally from the crumpled polymer organization. Simple crumpled polymer without loop extrusion and phase separation would also produce biphasic structures.

Yes, we agree. Also SR-EV leads to biphasic structures.

- L 448-449: any data to show that existing polymer modeling would predict a strong dependency of C_p(n) on the volumic fraction (in the range studied here)?

No, I don’t know a work predicting that.

- Fig. 4:- Large-scale structural properties (R^2(n) and C_p(n)) are not dependent on phi. Is it surprising that by construction, SR-EV only relaxes the system locally after SRRW application?

Excluded volume is considered at all length scales. However, as the decreasing C_p curves observed in theories and experiments imply, the fraction of overlap (or contacts) is more important at small separations (local) than at large separations. Yet, it was a surprise for us to observed negligible effect on phi.

- Why not make a quantitative comparison between predicted and measured C_p(n)? Or at least plotting them on the same panel.

Panels B and C are in the same scale and show a good agreement between SR-EV and experiments. However, it is not perfectly quantitative agreement. SR-EV represents the generic structure of chromatin and perfect agreement should not be expected.

- Comparison with an average C_p(n) over all the chromosomes would be better.

Possibly, but we don’t think it adds anything to the paper.

- In Figure 5,6,7 (and related text): authors often describe some parameter values that are 'closest to experiment findings'. Can the authors quantify/justify this? The various 'closest' parameters are different. Can the authors comment?

The folding parameter and average volume fraction are chose so that the agreement is best with the displayed experimental system, different cell for each case.

- Figure 5: why not show the experimental distribution from Ou et al?- Figure 6 & 7: experimental results. Can the authors show images from their own experiments? Can they show that cohesion/RAD21 is really depleted after auxin treatment?

It is currently under review in a different journal.

- In the Discussion, a fair discussion on the limitations of the methods (dynamics, etc) is missing.Minor- Line 34-36: the logical relationship between this sentence and the ones before and after is very unclear.- Along the text, authors use the term 'connectivity' to describe 3D (Hi-C) contacts between different regions of the same chromosome/polymer. This is misleading as connectivity in polymer physics describes the connection along the polymer and not in the 3D space.

No. I don’t think we used connectivity in that sense. We agree with your statement on the use of connectivity in polymer physics, and is what we always had in mind for this model.

- Line 92: typo.- On the SR-EV method: does the relaxation process create local knots in the structure?

We have not checked for knots.

- Table 1: the good correspondence with linker length is remarkable but likely 'fortunate', other chosen resolutions would have led to other results. Moreover, the model cannot account for the fine structure of chromatin fiber. Can the authors comment on that?

Fortunate to the extent that we sample the model parameter to overall catch the structure of chromatin.

- Line 211: 'without the need of imposing any parameter': alpha is a parameter, no?

Correct. Phrase deleted.

- L267-269 & 450-451: actually in Liu & Dekker, they do observe an effect on Hi-C map (C_p(n)), weak but significant and not negligible.

Our statements read ‘minimal’ and ‘relatively insensitive’. It is observed, but very small.

- L283-286: This is a perspective statement that should be in the discussion.

Moved to the Discussion, as suggested.

- L239-241: The authors seem to emphasize some contradictions with recent results on phase separation. This is unclear and should be relocated to discussion.

We just pointed out recent experiments, as stated. No intention to generate a discussion with any of them.

- L311-313: Unclear statement.- L316-325: This is not results but discussion/speculation.

Moved to Discussion

- Along the text: 'promotor'-> 'promoter'.

- Corrected.

- L364: explain more in detail PWS microscopy.
**Reviewer #2 (Recommendations For The Authors):**
Even though there are claims about nucleosome-resolution chromatin polymer, it is not clear that this work can generate structures with known nucleosome-resolution features. Nucleosome-level structure is much beyond a random walk with excluded volume and is driven by specific interactions. The authors should clarify this.